# A subterranean adaptive radiation of amphipods in Europe

Špela Borko [1✉], Peter Trontelj [1], Ole Seehausen [2,3], Ajda Moškrič [1,4] & Cene Fišer[1]

Adaptive radiations are bursts of evolutionary species diversification that have contributed to much of the species diversity on Earth. An exception is modern Europe, where descendants of ancient adaptive radiations went extinct, and extant adaptive radiations are small, recent and narrowly confined. However, not all legacy of old radiations has been lost. Subterranean environments, which are dark and food-deprived, yet buffered from climate change, have preserved ancient lineages. Here we provide evidence of an entirely subterranean adaptive radiation of the amphipod genus *Niphargus*, counting hundreds of species. Our modelling of lineage diversification and evolution of morphological and ecological traits using a time-calibrated multilocus phylogeny suggests a major adaptive radiation, comprised of multiple subordinate adaptive radiations. Their spatio-temporal origin coincides with the uplift of carbonate massifs in South-Eastern Europe 15 million years ago. Emerging subterranean environments likely provided unoccupied, predator-free space, constituting ecological opportunity, a key trigger of adaptive radiation. This discovery sheds new light on the bio-diversity of Europe.

[1] SubBio Lab, Biotechnical Faculty, University of Ljubljana, Ljubljana, Slovenia. [2] Aquatic Ecology and Evolution, Institute of Ecology and Evolution, University of Bern, Bern, Switzerland. [3] Department of Fish Ecology and Evolution, Centre for Ecology, Evolution, and Biogeochemistry, Swiss Federal Institute of Aquatic Science and Technology (EAWAG), Kastanienbaum, Switzerland. [4] Agricultural institute of Slovenia, Ljubljana, Slovenia. ✉email: spela.borko@bf.uni-lj.si

Adaptive radiations are large bursts of speciation from single ancestors, accompanied by extensive ecological diversification[1–5]. They have played a significant role in the origin of extant diversity in many major clades across the globe[3,5,6]. Europe, after Antarctica, stands out at the low end of global biological diversity[7,8]. This part of the world, whose natural history has been most thoroughly explored, has shown little evidence of extensive adaptive radiations. Most modern species assemblages of continental Europe seem to have assembled by immigration from elsewhere rather than by in situ adaptive radiations, with minor exceptions among flowering plants, butterflies, and fish in large lakes[9,10]. Albeit interesting because of their high rates of speciation, most of these are geographically narrowly confined and limited in the number of species[11,12]. Notwithstanding the lack of large extant faunal radiations, the European landmass contributed opulently to global species richness during Earth's deeper geological epochs. Fossil records and reconstructions suggest that European adaptive radiations unfolded between Eocene and Miocene[4,13], but these taxa either moved south- and eastward or went extinct due to tectonic changes and paleoclimate perturbations[4,10,13]. The temporary disappearance of the Mediterranean Sea 6–5 million years ago (Mya)[14] and the desiccation of paleo lakes[4] likely eliminated most descendants of older marine and lacustrine adaptive radiations.

There is scattered evidence suggesting that the legacy of Europe's prolific pre-Pleistocene faunal history is not entirely lost but that some groups have survived in subterranean environments that were isolated and protected from the turbulent geological and paleoclimatic history. Indeed, the global species richness of caves and groundwater is the highest in European karstic areas of the Pyrenees, Southern Alps, and the Dinaric Karst. This pattern is well-documented but insufficiently understood[15,16]. A second hint comes from the finding that even environments extremely poor in energy and ecological variation such as deep karstic caves can support considerable adaptive evolutionary diversification[17]. Finally, modern speleological techniques have enabled an advance of biological knowledge in a habitat that is as challenging to explore as the deep sea[18]. Morphological and molecular data from thousands of individuals and hundreds of new species is now available for several subterranean taxa[19–21]. The front-runner of them all is the subterranean amphipod genus *Niphargus*, the largest genus of freshwater amphipod crustaceans in the world[16,22,23].

Here, we present and test an entirely new view on the evolution of subterranean biodiversity and on the origins of major extant European faunal components. We do so by demonstrating that *Niphargus* is a mega-diverse genus with hundreds of species that has not only evolved and diversified entirely on the European continent but has survived here from the times when the landmasses emerged from the sea (Fig. 1). We analyse the tempo and mode of diversification and ecological disparification of this exclusively subterranean clade and show that diversification patterns satisfy the predictions of evolution by adaptive radiation. Next, using spatio-temporal correlations between diversification events and the geological uplift of large carbonate massifs, we suggest that the formation of caves and subterranean habitats created a multitude of ecological opportunities, the quintessential condition for adaptive radiation[24].

## Results

### Mega-diversification of *Niphargus*.
We reconstructed the most complete multilocus phylogeny of *Niphargus*[25] (Fig. 2), containing more than twice as many species as the next largest subterranean clade analysed so far[19]. The analyses of 377 Molecular Operational Taxonomic Units (MOTUs) are based on DNA

sequences of one mitochondrial and seven nuclear genes (7067 bp in total, with a mean value per MOTU 3017 ± 2146 (standard deviation)) (Supplementary Data 1). MOTUs also include species discovered in recent studies but not formally named yet (see 'Methods' section).

We found that the genus *Niphargus* originated in the Middle Eocene (mean value 47 Mya, highest posterior density (HPD) interval 56–39 Mya) in the region that is contemporary Western Europe (Fig. 3 and Supplementary Data 2, Supplementary Fig. 1). Lineage through time analysis[26] identified an increase in net diversification rate that occurred at ~15 Mya with subsequent slowdown starting at 5 Mya (Fig. 2a), implying dynamics consistent with adaptive radiations, a so-called 'early burst' ($\gamma = -5.1719$, $p$-value = 0). Testing alternative models of diversification, we identified an increase in diversification between 15 and 16 Mya corresponding to increased speciation (best model), or increased speciation together with increased carrying capacity (suboptimal model), rather than decreased extinction rates. The impact of extinction is inferred to be negligible in all models (Table 1).

*Niphargus* species fall into six broad categories by aquatic subterranean habitat: (1) unsaturated fissure system, (2) interstitial groundwater, (3) cave lakes, (4) cave streams, (5) shallow subterranean[27] and (6) groundwater with specific chemistry (brackish, sulfidic, acidic, or mineral) (Supplementary Data 3). The reconstruction of ancestral states using stochastic habitat mapping under Brownian motion suggested that during the first 20–30 million years *Niphargus* species predominantly lived in and dispersed via interstitial coastal and alluvial habitats (interstitial groundwater and shallow subterranean habitat categories). This period was followed by colonisation and adaptation to new subterranean habitats that took place repeatedly in different clades (Figs. 2, 4 and Supplementary Fig. 2). The ecological diversification was analysed using changes through time plots (CTT), which estimates the mean number of character changes per sum of branch lengths in a given period and compares it against a null model of constant evolution[26]. The CTT plot suggested ecological diversification consistent with a neutral scenario at the onset, followed by a substantial drop, and a late phase that begins with a sudden steep increase of ecological diversification at 15 Mya and further increase to the present (Fig. 2). The onset of increased diversification agrees in time with the onset of increased speciation rates. This suggests that ecological diversification and speciation took place hand in hand, coinciding with the orogenesis and karstification of South-Eastern Europe in the Early Miocene (see section 'Geographical origin of adaptive radiations').

The functional morphology of *Niphargus* species (body size, appendage length, body shape) corresponds to environmental conditions to which these species are adapted and can be used to predict components of the ecological niche such as water flow, size of subterranean spaces, mode of locomotion and to some extent feeding habits[17,28]. There is no simple one-to-one correspondence between the morphotypes (Fig. 1) and the habitats listed above. Different morphotypes sometimes share the same habitat but may differ in their trophic niches. Such examples are species of different body size co-inhabiting cave lakes[17], or species of different body shapes coexisting in interstitial groundwater[23] (see Fig. 2b for an example). Conversely, some species with generalist morphology can be found in more than one subterranean habitat[29].

In the next step, we explored the evolutionary dynamics of morphological disparity within *Niphargus*, using disparity through time plots (DTT)[30]. Eleven morphological quantitative traits served as a proxy for the ecological function of species[17,23] (Supplementary Data 4). The DTT approach enables the investigation of

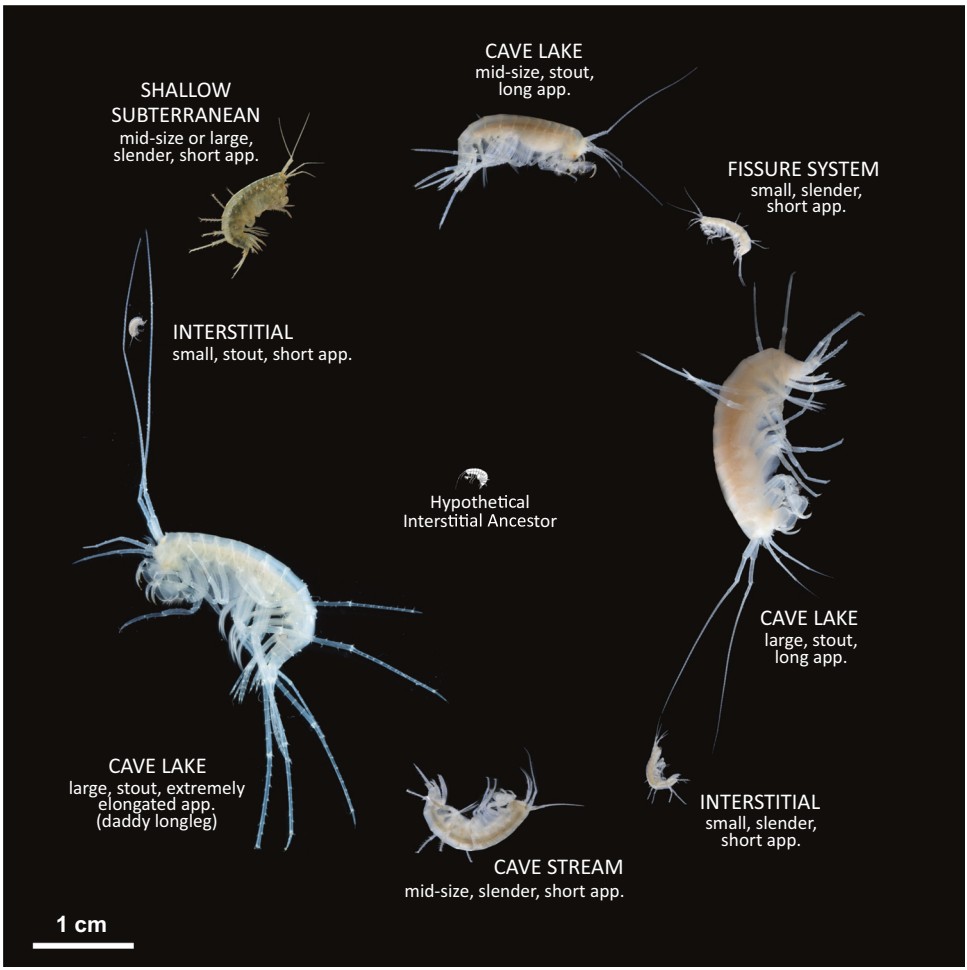

**Fig. 1 Morphological and ecological diversity of *Niphargus* species.** Adaptive radiation of *Niphargus* produced several morphotypes that inhabit distinct subterranean habitats and niches. Different-sized morphotypes can occur together in cave lakes, and differently shaped morphotypes can occur together in interstitial habitats[17]. These morphotypes evolved in at least five adaptive radiation events from hypothetical small-bodied ancestors from the shallow interstitial habitat[20,23]. Appendages are abbreviated as *app*. Photographs used with author's permission: Denis Copilaş-Ciocianu (shallow subterranean), Teo Delić (all others).

disparity patterns in conjunction with clade age. For each divergent event (i.e. each node) we calculated the average of the disparities of all subclades whose ancestral lineages were present at that time, standardised against the overall disparity. Disparity values near 0 imply that subclades contain relatively little variation present within the entire clade, and most variation is partitioned as differences between subclades. Values near 1 imply that subclades contain a substantial proportion of the total variation, indicating that species within subclades have independently evolved to occupy similar regions of the ecomorphological space[30,31]. The observed DTT curve was compared to the null hypothesis of neutral evolution of morphology in which we simulated the relative disparities obtained from Brownian motion model[30]. The DTT plot suggested an early burst of disparity when the two major lineages arose ~35 million years ago, followed by 15 million years of continuous disparity decline that is in accord with neutral models of evolution. This decline is sharply ended by a significant recovery of morphological disparity at 15 Mya, pointing towards independent diversification of ecological traits within subclades. This phase of phenotypic diversification coincides with the increase in the rates of species accumulation and their ecological diversification (Fig. 2: LTT and CTT). The morphological disparity index (MDI)[31] that measures the overall difference in the relative

disparity of a clade against the disparity expected under the null model, showed insignificantly higher disparity than expected by the null model (MDI = 0.028, $p$ = 0.4). We attribute this insignificant result to the overall DTT dynamics where low disparities in early history cancelled out higher values during the last 15 Mya. The rank envelope test that tests how extreme the reconstructed DTT curve is compared to the simulated curves, showed that the DTT curve falls outside the expected 95% confidence intervals of the null model simulations ($p$-interval 0.0009–0.0270). Visual inspection of the DTT plot (Fig. 2) showed that this deviation happened from 15 Mya onward. We also tested whether the evolutionary rates of the eleven studied traits changed in time using the univariate node height test[31,32]. The results were significant for all eleven traits, showing that the rate of their evolution indeed systematically increased during the evolutionary history of the genus (Supplementary Table 1 and Supplementary Fig. 3). Further exploration of multivariate evolutionary models applied to morphological traits[33] suggested that morphological diversification of the genus can be best explained by a switch from the neutral Brownian motion model of morphological diversification to an early burst model at the time of presumed increase in diversification (15.5 Mya). This shift suggests increased morphological diversification that finally slowed down again at 2 Mya, possibly reflecting the emergence of

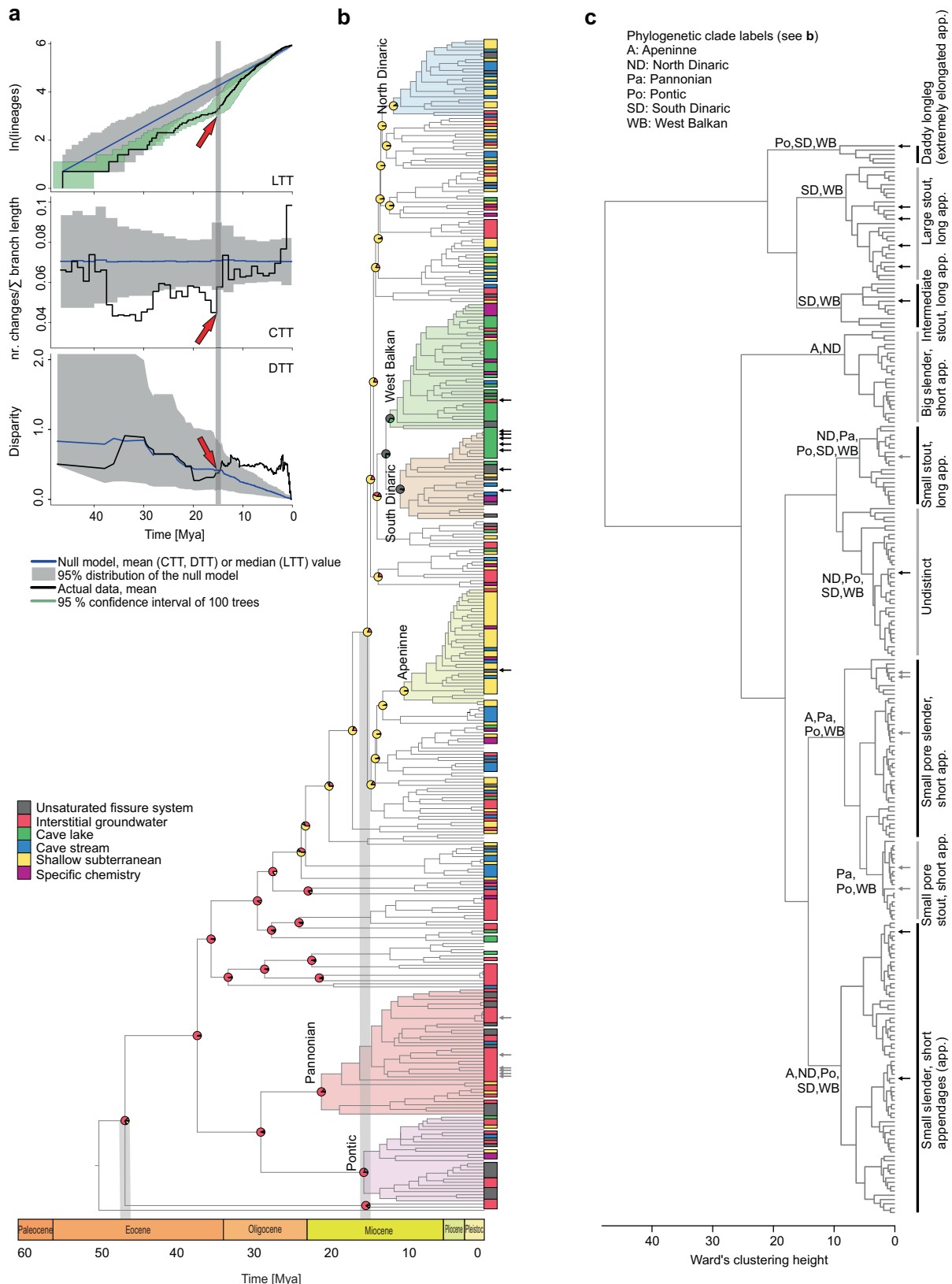

ecological opportunity associated with the formation of karstic massifs, and the subsequent filling of the new ecological niches as adaptive radiation progressed (Table 2).

In summary, all diversification analyses suggested sudden increases in the rates of species accumulation, morphological evolution and ecological disparification at ~15 Mya—the time when novel habitat emerged at a grand scale[27,34,35] (Fig. 3). These results are consistent with predictions from the ecological theory of adaptive radiation, whereby the emergence of a large volume of novel and diverse habitat triggers the evolutionary diversification of a lineage suited to this habitat.

**Fig. 2 Three different aspects of the adaptive radiation of *Niphargus*. a** Lineage diversification, ecological and morphological disparification notably accelerated ~15 Mya. The pattern is well visible on LTT, CTT and DTT plots. The time of an evolutionary shift is indicated by red arrows and grey line. **b** Calibrated phylogeny, species' habitats and ancestral habitat reconstructions of *Niphargus*. Tips are labelled according to extant habitat. Pies on selected nodes represent reconstructed ancestral habitat (for complete reconstructions see Supplementary Fig. 2). Clades that were subjected to further analyses are coloured and named. **c** Cluster dendrogram based on eleven functional morphological traits. The same groups were partially recovered by PCA (Supplementary Fig. 8). The phylogenetic composition of morphological groups is labelled on the dendrogram. Cluster analysis shows that unrelated species evolved similar morphology (see clade's acronyms on basal nodes of dendrogram). High morphological disparity presumably allowed high levels of syntopy and the formation of species-rich communities of closely related species. As an example, black and grey arrows on **b** and **c** point to species from a cave (Vjetrenica Cave System, Bosnia and Hercegovina) and interstitial (Sava river close to Medvode, Slovenia) communities, respectively. Note that many community members are closely related (same clade) but belong to different morphological clusters.

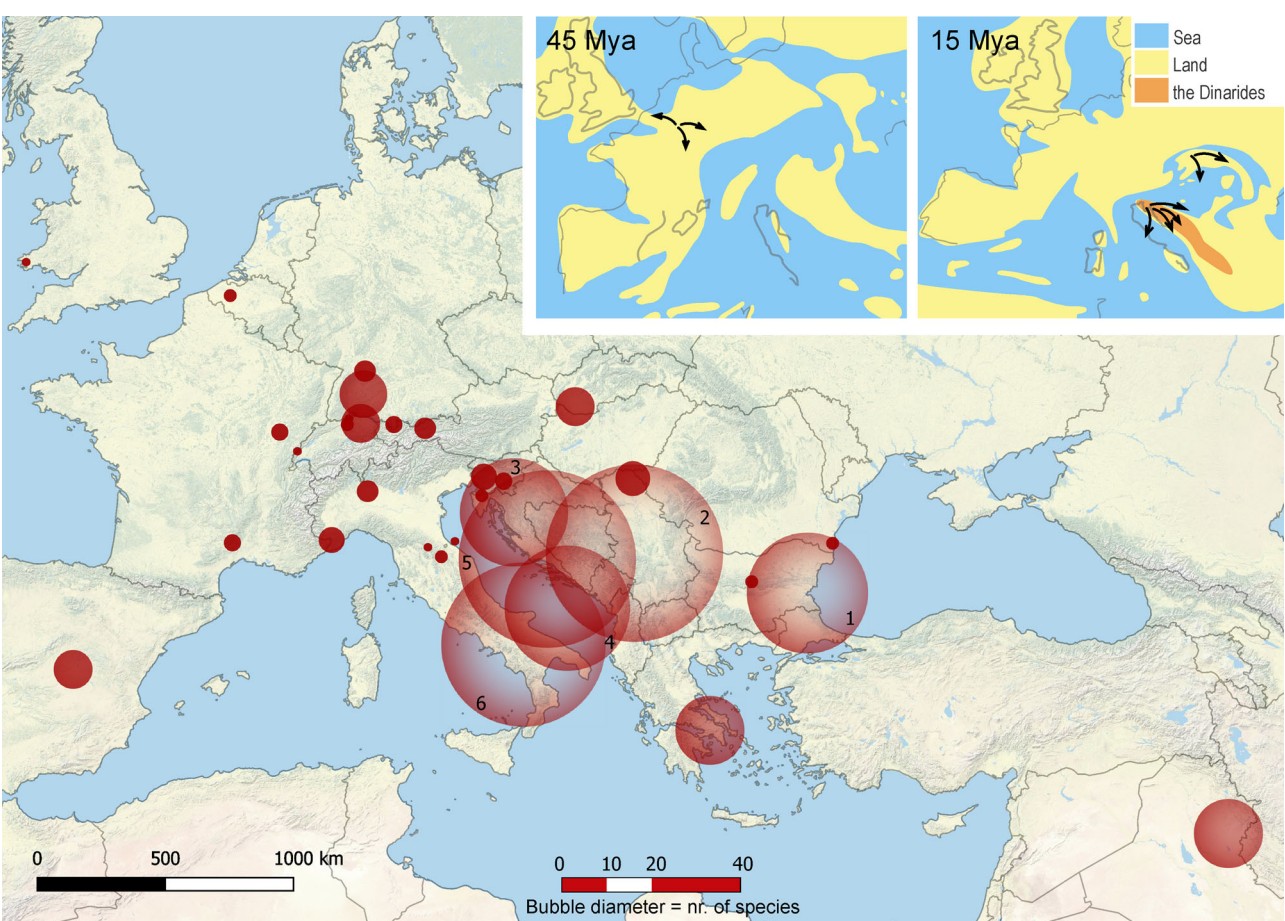

**Fig. 3 Geographical extent of the adaptive radiation of *Niphargus*.** Bubbles represent the geographic position of subclades; their size reflects species numbers. Clades are numbered as follows: 1. Pontic, 2. Pannonian, 3. North Dinaric, 4. South Dinaric, 5. West Balkan and 6. Apennine clade. Inset: Phylogeny-based reconstruction of ancestral areas and directions of colonisation at 45 and 15 Mya, plotted on corresponding paleo maps, adapted after[39,40]. The geographic origin of species-rich subclades that arose through adaptive radiation corresponds to emerging karstic regions in South-Eastern Europe at about 15 Mya. The contemporary map was produced using QGIS[77] and Esri World Physical Map[78].

**Multiple independent radiations**. The genus *Niphargus* comprises several large, geographically well-defined clades. In exceptional settings, large adaptive radiations can be the sum of several independent radiations occurring in closely related lineages[3,36]. We selected six well-supported reciprocally monophyletic clades that were geographically well-defined and sufficiently large ($N \geq 25$ species) to be explored for patterns of adaptive radiation by repeating LTT, CTT and DTT analyses on each clade separately. They are distributed mostly in the karstic regions of Italy and South-Eastern Europe and spatially overlap to a various extent (Table 3, Fig. 2 and Supplementary Fig. 4). The Pontic and the Pannonian clade diverged from the rest of *Niphargus* early (38 Mya, HPD 40–35 Mya), and split 29 Mya (HPD 34–24 Mya). The South Dinaric, West Balkan, North

Dinaric and the Apennine clade are younger, they started to diversify at 15–11 Mya (Supplementary Fig. 1).

The analyses of diversification (LTT plots and $\gamma$-test) suggested that all clades display the pattern of an early burst of species accumulation with onset between 15 and 5 Mya. CTT plots did not deviate from null models, but DTT analyses imply adaptive radiation patterns in the Pontic, Pannonian, West Balkan and the North Dinaric radiations (Table 4 and Fig. 4). Dynamics of species accumulation and ecological disparification among these four clades differ (Table 4 and Fig. 4), possibly reflecting differences in habitat diversity among regions where these radiations unfolded or differences in the time of arrival of these lineages in the novel habitats. The Apennine clade is composed mainly of morphologically similar, still undescribed species.

**Table 1 Comparison of evolutionary models tested in order to identify the diversification pattern and rate shifts in the phylogeny of *Niphargus*.**

| Model | la1 | mu1 | K1 | la2 | mu2 | K2 | t-shift | AIC | AICw |
|---|---|---|---|---|---|---|---|---|---|
| Shift in speciation rate | 0.09 | 0 | 505.10 | 0.26 | mu_1 | K_1 | 15.70 | 2214.64 | 0.57 |
| Shift in speciation rate and carrying capacity | 0.12 | 0 | 45.15 | 0.25 | mu_1 | 523.01 | 15.52 | 2217.02 | 0.17 |
| Shift in speciation rate, extinction rate and carrying capacity | 0.14 | 0.01 | 35.20 | 0.26 | 0 | 501.18 | 15.70 | 2217.21 | 0.16 |
| Shift in extinction rate and carrying capacity | 0.24 | 0.07 | 25.08 | la_1 | 0 | 552.42 | 15.49 | 2217.97 | 0.11 |
| Shift in carrying capacity | 0.20 | 0 | 1.56 | la_1 | mu_1 | 666.39 | 37.51 | 2238.11 | 0 |
| Diversity dependent model without shift | 0.21 | 0 | 609.67 | / | / | / | / | 2241.14 | 0 |

Optimised parameters: la = speciation, mu = extinction, K = carrying capacity, t-shift = time of shift in Mya. The number denotes parameters before (1) or after (2) the shift. All optimising functions reached convergence.

Because of the lack of morphological data, we could not derive conclusions about the nature of this radiation.

Finally, we explored whether these clade-level radiations show signs of between- or within clade convergent evolution. We used SURFACE, a method that tests whether lineages have converged in phenotype without using a priori designations of ecomorphs[37]. It applies Hansen's model of adaptive peaks[38] modelled onto the phylogeny and assumes that in the case of convergence similar phenotypes in different clades could be attributed to the same adaptive peaks. Three models that best explained the evolution of ecomorphological traits predicted 14–16 different adaptive peaks, of which 11–12 were found to be convergent whereas only three to four were unique. These peaks partially correspond to clusters from the morphological analysis (Fig. 2). Of 11 convergent peaks, two evolved multiple times within one clade, whereas nine peaks evolved in two or more focal or non-focal clades. The results of the best model are illustrated in Supplementary Fig. 5.

We tentatively conclude that at least four out of six large and phylogenetically distinct clades could be considered as adaptive radiations. Although showing some degree of convergence, the radiations overall adapted to distinct sets of adaptive optima, especially so among the South-Eastern Europe clades.

**Geographic origin of adaptive radiations**. The increase in diversification and disparification around 15 Mya corresponds to the emergence of karstic regions in South-Eastern Europe in the area of the modern South-Eastern Alps, the Dinarides and the Carpathians[39,40]. This area underwent a complex geological history. The collision of the European and Adriatic slabs during the Eocene (66–34 Mya) caused vivid tectonic movements that resulted in drastic topographic changes[41]. In the Oligocene (32–23 Mya), the South-Eastern Alps and the Dinarides emerged from the surrounding seas, and the uplift of the Carpathians begun. The exposure of carbonate rocks of the Alpine-Carpathian-Dinaridic orogenic system to atmospheric processes triggered karstification, i.e. the formation of caves and a variety of other subterranean habitats[17,27]. This process began in the Oligocene and peaked in the Early Miocene (23–16 Mya). In that period, mountain uplift continued and the three mountain ranges acted as islands in the Parathethys Sea, occasionally connected during marine regressions. During the later Miocene (16–14 Mya), a mosaic of saline and freshwater aquatic environments and new emerging carbonate islands replaced the Parathethys, which completely regressed from Late Miocene onwards (11 Mya)[34,35].

The dynamic orogenesis of South-Eastern Europe opened a new unpopulated ecological space. These vast new freshwater environments, initially free of predators and competitors, constituted an ecological opportunity for lineages preadapted to freshwater subterranean lifestyles to colonise and undergo subterranean adaptive radiations. We tested this hypothesis by reconstructing ancestral habitats and ancestral areas of the six *Niphargus* clades at 20 and 15 Mya. The evolution of ancestral ranges reconstructed under the Brownian motion model within a Bayesian framework suggested that *Niphargus* originated in what is now a part of Western Europe (Fig. 3). According to ancestral habitat reconstruction these species lived in and dispersed via interstitial and shallow subterranean water systems to South-Eastern Europe. The ancestral ranges of the six clades coincide with emerging carbonate islands of the Tethys Ocean and later Parathethys Sea, nowadays representing the South-Eastern Alps, the Dinaric Karst, and parts of the Carpathian arch (Fig. 3 and Supplementary Data 2).

These results suggest that Miocene karstification on island-like parts of the continent apparently provided an ecological opportunity for groundwater-dwelling *Niphargus* that was able to colonise and subsequently exploit the subterranean realm. The newly colonised empty space possibly triggered ecological divergence from the ancestral interstitial and shallow subterranean forms, followed by morphological diversification and speciation within each independent radiation (Table 3).

## Discussion

The recovered patterns are concordant with predictions of adaptive radiation theory: a lineage with heritable ecological versatility to diversify that finds its ecological opportunity, will rapidly proliferate into ecologically different species, until the available niches fill up[5]. We applied different methods which conservatively pinpointed that tree-wide shifts in evolutionary dynamics spatially and temporally correspond to the onset of ecological opportunity.

*Niphargus* originated from marine ancestors[20] 56–39 Mya, in Western Europe as an interstitial amphipod. For the first 20–30 million years, speciation and ecological diversification followed or even lagged behind neutral expectations. The genus presumably dispersed via coastal or brackish interstitial aquatic habitats (Fig. 2) and accumulated genetic variation, which supported rapid speciation that followed[2].

Uplift of several carbonate platforms in South-Eastern Europe as islands in the Parathethys and subsequent karstification in the Early Miocene, 23–16 Mya, created new, ecologically diverse subterranean habitats. *Niphargus* lineages colonised these new predators- and competitor-free habitats and underwent several evolutionary radiations. At least five of these radiations exhibit patterns of lineage diversification and trait evolution consistent with adaptive radiation, and the available information from the undersampled clades in Greece and Iran[42] hint there may be more such clades. Early habitat diversification, detected in tree-wide CTT analysis but not at the level of an individual clade, may have constrained further clade-level morphological

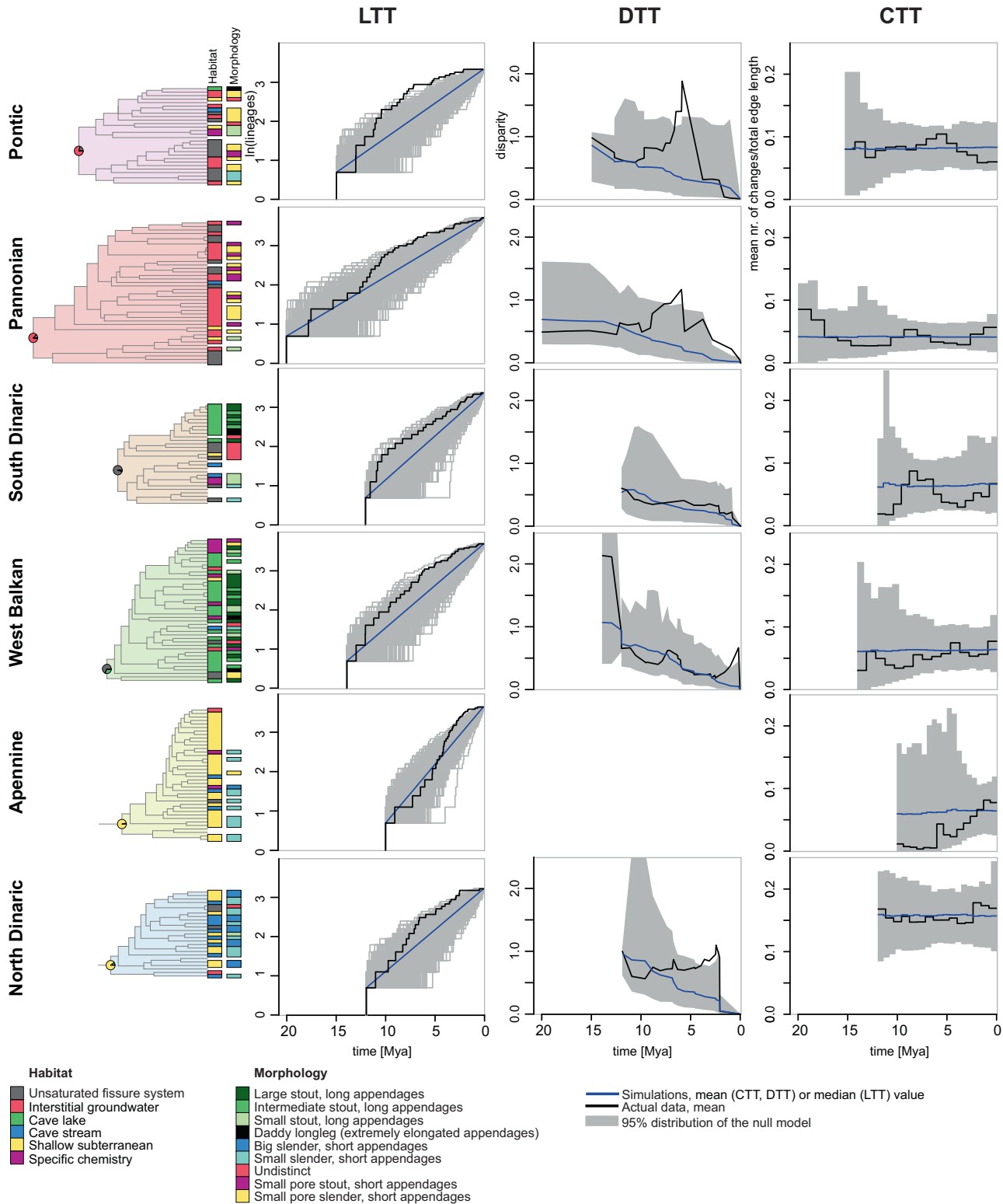

**Fig. 4 Evolution of six major *Niphargus* subclades.** Zoomed phylogenies with plotted habitats and morphotypes (missing data not available), lineage through time (LTT), changes through time (CTT) and disparity through time (DTT) plots. The DTT analysis could not be performed for the Apennine clade due to too much missing data.

diversification. Subsequent morphological diversification of clades predominantly unfolded within one or few habitat types, like cave lakes (West Balkan and South Dinaric clade) or interstitial groundwater (Pontic and Pannonian clade). This within- and between-habitat diversification prompted high levels of sympatry

and allowed for uniquely species-rich communities, counting up to nine species per site[17,23] (Fig. 3). High local diversity combined with multiple independent radiations can explain the extraordinarily high number of species in the Dinaric Karst, a global subterranean biodiversity hotspot[16]. The mega adaptive radiation

**Table 2 Comparison of multivariate evolutionary models with or without shift, tested to identify the morphological diversification pattern of the radiation of *Niphargus*.**

| Model | logLik | params | AIC | diff | AICw |
|---|---|---|---|---|---|
| Brownian motion to early burst with independent drift | −1420 | 9 | 2859 | 0 | 0.555 |
| Ornstein–Uhlenbeck to early burst | −1422 | 9 | 2862 | 2.78 | 0.138 |
| Brownian motion to Ornstein–Uhlenbeck with independent drift | −1420 | 11 | 2862 | 3.02 | 0.123 |
| Brownian motion to early burst | −1425 | 6 | 2862 | 3.36 | 0.104 |
| Early burst to Ornstein–Uhlenbeck with independent drift | −1420 | 12 | 2864 | 5.02 | 0.045 |
| Ornstein–Uhlenbeck to early burst with independent drift | −1420 | 12 | 2865 | 5.88 | 0.029 |
| Ornstein–Uhlenbeck | −1426 | 8 | 2868 | 9.4 | 0.005 |
| Early burst to Ornstein–Uhlenbeck | −1427 | 9 | 2871 | 12.31 | 0.001 |
| Brownian motion to Ornstein–Uhlenbeck | −1454 | 8 | 2925 | 65.8 | 0 |
| Early burst to Brownian motion | −1490 | 6 | 2991 | 132.6 | 0 |
| Ornstein–Uhlenbeck to Brownian motion | −1489 | 8 | 2994 | 134.8 | 0 |
| Early burst to Brownian motion with independent drift | −1489 | 9 | 2996 | 137.5 | 0 |
| Ornstein–Uhlenbeck to Brownian motion with independent drift | −1890.18 | 21 | 3822.36 | 250.55 | 0 |
| Brownian motion | −1915.91 | 9 | 3849.82 | 278.02 | 0 |
| Early burst | −1915.91 | 10 | 3851.82 | 280.02 | 0 |

**Table 3 Summary of diversification patterns of the genus *Niphargus* and its major subclades.**

| Clade | Nr. of species analysed | Distribution | Nr. of habitats occupied | Nr. of morphotypes | Diversification pattern |
|---|---|---|---|---|---|
| *Niphargus* genus | 377 | From Ireland to Iran | 6 | 9 | Adaptive radiation |
| Pontic | 28 | Eastern Europe to Crimean Peninsula | 6 | 6 | Adaptive radiation |
| Pannonian | 41 | Northern Italy, Pannonian Basin and Carpathian arch | 4 | 3 | Adaptive radiation |
| South Dinaric | 29 | Southern Dinaric Region, Adriatic Islands, southern parts of Apennine Peninsula | 5 | 6 | Adaptive radiation |
| West Balkan | 41 | North-Western Balkan | 6 | 8 | Adaptive radiation |
| Apennine | 38 | Apennine Peninsula, Eastern coasts of Adriatic Sea | 5 | 3 | Non-adaptive radiation |
| North Dinaric | 25 | Northern Dinaric Region, Adriatic Islands, central parts of Apennine Peninsula | 4 | 4 | Adaptive radiation |

**Table 4 Lineage through time (LTT), changes through time (CTT) and disparity through time (DTT) statistics for six major Niphargus subclades.**

| Clade | LTT statistics | DTT statistics (MDI; p-interval of ranked envelope test) | CTT |
|---|---|---|---|
| Pontic | $\gamma = -3.9$, $p$-value $= 0$ | 0.15; 0.001–0.025 | Random |
| Pannonian | $\gamma = -2.6$, $p$-value $= 0.009$ | 0.17; 0.001–0.026 | Random |
| South Dinaric | $\gamma = -1.8$, $p$-value $= 0.076$ | 0.06; 0.364–0.371 | Random |
| West Balkan | $\gamma = -3.1$, $p$-value $= 0.002$ | 0.14, 0.001–0.033 | Random |
| Apennine | $\gamma = -3.9$, $p$-value $= 0$ | n.a.[a] | Recent increase |
| North Dinaric | $\gamma = -2.9$, $p$-value $= 0.004$ | 0.15; 0.001–0.028 | Random |

[a]The DTT analysis could not be accurately performed for the Apennine clade due to missing data.

of the amphipod genus *Niphargus* that generated 20–25% of all freshwater amphipods in the World, can be seen as an exception to the widespread pattern where the fauna of present-day Europe contributes relatively little to global biological diversity[8,43].

At a first glance, it seems paradoxical that what might be the largest surviving pre-Pleistocene adaptive radiation on the European continent took place in an environment as desolate as the karstic underground. The environment is ecologically simple, permanently dark, and nutrient-deprived[27]. As it lacks primary producers, ecological opportunity—an environmental prerequisite for adaptive radiation[24]—does seem unlikely to be fulfilled. However, adaptive radiations have been reported from other oligotrophic habitats, including the abyssal zone of deep lakes[44], the deep sea[45] and the Antarctic[46]. Along with studies from major

clades in the tree of life and large biomes[1,3], these studies support Simpson's idea that adaptive radiations are a universal phenomenon and an important global generator of biodiversity[5].

We noted a pattern of independent regional adaptive radiations in several clades. This mechanism likely contributed to the outstanding species richness of *Niphargus*, across different geographic scales[22]. Consequently, the *Niphargus* radiation is similar in size to the radiations of amphipods from the Baikal lake[44,47]. Evolution of distinct phenotypes in independent adaptive radiations as already reported in plants and vertebrates[1,3,48] may be more common than previously thought, yet overlooked in studies that were narrow in taxonomic focus[49].

In the Western Palearctic, ancient in situ adaptive radiations that predate the climate chaos of the Pleistocene do not reveal

themselves readily. The case of *Niphargus* suggests that one must search for them in environments that are insulated against the effects of climate fluctuations. Climate history clearly had a strong impact on the rise and loss of biodiversity in Europe[4]. Climatic perturbations wiped out faunas that had evolved in ancient adaptive radiations[4]. Notably, these Pleistocene extinctions created ecological opportunity[24] for multiple Holocene adaptive radiations of limited extent[11,12]. Only in ecosystems that were shaded from the devastating effects of climatic fluctuations could the full historical richness of older radiations be preserved. If this hypothesis is correct, further relic adaptive radiations should be expected among taxa from environments that were so far neglected, such as the subterranean realm and deep soil[21,50]. Whether or not we shall be able to discover them depends also on the progression of adverse anthropogenic impacts: while resilient and buffered from ecological fluctuations throughout geological history, these species are not immune to habitat destruction and may be particularly vulnerable to pollution[51].

## Methods

**Dataset**. The molecular dataset consisted of all available genetic data for described and undescribed *Niphargus sensu lato* species. Several species formally included in different niphargid genera (*Carinurella, Haploginglymus, Chaetoniphargus, Niphargobates*) are nested within the *Niphargus* radiation. Some of them are monophyletic, and others are not. Although the monophyly of *Niphargus sensu lato* was established multiple times[25,29,52] the nested genera were never formally synonymized, and thus we use the valid nomenclature. Sequences included newly acquired and previously unpublished sequences (SubBio Database[53], accessed on 20. 2. 2018) and existing GenBank records[54]. To account for cryptic species, we substituted nominal species with MOTUs, molecularly delimited in the previous works[25,29,55,56]. Most of these species were delimited using multilocus species delimitations; only a smaller fraction was delimited using only mitochondrial COI gene sequences. In these cases, we relied on delimitations using the most conservative delimitation approach, i.e. a 16% patristic distance treshold[20]. In total, our dataset counted 377 MOTUs of *Niphargus*, rooted with five outgroup species from the genera *Niphargellus, Microniphargus* and *Pseudoniphargus*[57]. Of these, 223 MOTUs corresponded to nominal species (50% of the 447 nominal *Niphargus* species (WORMS[58])); 154 MOTUs are not yet formally named.

Ecological information on the studied species was extracted from literature compiled in the European Groundwater Crustacean Database (EGCD)[16], from own data and from databased field notes (SubBio Database[53], accessed on 20. 2. 2018). We assigned 331 species to six habitat categories ((1) unsaturated fissure system, (2) interstitial, (3) cave lakes, (4) cave streams, (5) shallow subterranean and (6) groundwater with specific chemistry (brackish, sulfidic, acidic or mineral)). Species for which ecological data are lacking and MOTUs identified from published GenBank sequences were left uncharacterised. Morphometric traits included 11 morphological continuous traits known to correspond to habitat properties[17,23]. The traits were measured using standard protocol[59]. Briefly, we partially dissected the specimens, mounted them on glycerol slides, and photographed them using a ColorView III camera mounted on an Olympus SZX stereomicroscope. We measured the photographs using the programme CellB (Olympus, 2008) to the precision of 0.01 mm. Depending on the availability of material, we measured between 1 and 20 individuals per species. In many species, samples were heavily damaged. In such cases, we supplemented the measurements with information available in species descriptions. For 26 species we had no appropriate samples, but we could extract measurements from the original descriptions. In total, we compiled a morphometric dataset for 256 species. Ethical issues are not applicable to this study.

**DNA sequences**. Genomic DNA was extracted using the GenElute Mammalian Genomic DNA Miniprep Kit (Sigma-Aldrich, United States). We amplified eight phylogenetically informative markers: the mitochondrial cytochrome oxidase I (COI) gene, two segments of the 28S rRNA gene, the histone H3 gene (H3), a part of the 18S rRNA gene, as well as partial sequences of the phosphoenolpyruvate carboxykinase (PEPCK), glutamyl-prolyl-tRNA synthetase gene (EPRS), heat shock protein 70 (HSP70) and arginine kinase (ArgKin) genes. Oligonucleotide primers and amplification protocols are given in Supplementary Table 4. Markers that were sequenced from several specimens of the same MOTU are marked as chimaeras in Supplementary Data 1. To exclude misidentification, chimeric specimens were from the same or nearby localities with identical sequences of overlapping markers of high resolution (e.g. COI). In the dataset, 301 MOTUs were represented by at least two fragments. For 76 MOTUs only the COI fragment was available.

Nucleotide sequences were obtained by the Macrogen Europe laboratory (Amsterdam, Netherlands) using amplification primers and bi-directional Sanger sequencing. We edited and assembled chromatograms in Geneious 11.0.3. (Biomatters Ltd, New Zealand).

**Phylogeny and molecular clock**. The sequences were aligned in Geneious with MAFFT v7.388[60] plugin, using E-INS-i algorithm with scoring matrix 1PAM/k = 2 with the highest gap penalty. Alignments were concatenated in Geneious. We used Gblocks[61] to remove gap-rich regions from the alignments of non-coding markers prior to phylogenetic analysis, with settings for less stringent selection.

We used Partition Finder 2 for determining the optimal substitution model for each partition[62,63] under the corrected Akaike information criterion (AICc). The selected optimal substitution models are listed in Supplementary Table 5.

Backbone phylogenies (301 *Niphargus* MOTUs with at least two fragments and five outgroups MOTUs) were reconstructed using Bayesian inference (BI) and maximum-likelihood (ML) in MrBayes 3.2.6[64] and IQ-TREE 1.6.6[65], respectively (Supplementary Fig. 6 and 7). In MrBayes we run two simultaneous independent runs with eight chains each for 200 million generations, sampled every 20,000th generation. We discarded the first 25% of trees as burn-in and calculated the 50% majority-rule consensus tree. Convergence was assessed through average standard deviation of split frequencies, LnL trace plots, potential scale reduction factor (PSRF), and the effective sample size. Results were analysed in Tracer 1.7[66]. The ML tree was calculated in IQ-TREE using the same evolutionary models as in BI, with ultrafast bootstrap approximation (UFBoot)[67], H-like approximate likelihood ratio test[62] and -bnni function to reduce the risk of overestimating branch supports with UFBoot. Phylogenetic analyses were run on the CIPRES Science Gateway (http://www.phylo.org). For the molecular clock analysis, we built a time-calibrated chronogram of all 377 *Niphargus* MOTUs with BEAST 2[68] (Supplementary Fig. 1).

We set four internal calibration points. (1) Fossil: 'modern-looking' *Niphargus* fossils from Eocene Baltic amber estimated to be 40–50 million years old[69]. Their morphology resembles modern *Niphargus* species with distinct pectinate dactyls. We placed the calibration point on the node where this particular morphological character occurs for the first time (*N. ladmiraulti*). (2) European niphargids: the final submergence of the land bridges between Eurasia and North America presumably prevented the dispersal of niphargids towards Greenland and North America. If so, niphargids are most probably younger than the three natural bridges connecting North Europe, Greenland and North America, which existed between 57 and 71 Mya[70]. (3) Crete clade: species from Crete form a highly supported monophyletic group with closest relatives on mainland Greece. The calibration point relies on the isolation of Crete Island from the mainland that happened at the end of the Messinian salinity crisis (5.3–5 Mya)[71]. (4) Middle East clade: species from Lebanon to Iran form a highly supported monophyletic group of eastern *Niphargus*. With rising sea levels and the opening of the connection between Paratethys and the Mediterranean basin 11–7 Mya, the Dinarides and eastern mainland became separated[39,40]. Technical details of calibration points are given in Supplementary Table 6.

To assess the degree of incongruence among the calibration points, we run analyses using each calibration point separately and compared age distributions for main nodes (Supplementary Table 2). For each gene partition we used a set of priors as followed: linked birth-death tree model, unlinked site models as in previous analyses, with fixed mean substitution rate and relaxed clock log-normal distribution with estimated clock rate. We used default settings of distributions of all estimates. We run the analyses for 200 million generations, sampled every 20,000 generations. The first 25% of trees were discarded as burn-in.

All subsequent analyses were run on the clock-calibrated trees of *Niphargus* (377 MOTUs). To account for phylogenetic uncertainty, we used a random subset of 100 calibrated trees and maximum clade credibility (MCC) tree. To assess the potential influence of missing sequence data in the dataset, we repeated the BEAST 2 and through time analyses (LTT, CTT, DTT and subsequent statistics) on the subset of 301 *Niphargus* MOTUs with at least two markers (Supplementary Fig. 8). The results were congruent with the results obtained from the extended dataset and showed a negligible effect of missing data on the final phylogeny reconstruction and downstream analyses.

**General strategy in diversification analyses**. The theory of adaptive radiation predicts an initial burst of speciation and disparification rates followed by a decline as the available niches fill. We tested this prediction using two alternative strategies. First, we analysed species diversification and ecological as well as morphological disparification through time against simulated null models. Secondly, we tested which evolutionary models of diversification and morphological disparification best fit to our data.

The analyses were run on the whole dataset and repeated on six selected clades that satisfied three conditions: high node support, a high number (>25) of species and geographically well-defined distribution within limited areas (Pontic, Pannonian, North Dinaric, South Dinaric, West Balkan and Apennine clade). MOTUs with missing morphometric data were excluded from the DTT and multivariate modelling analyses. The clades are marked on the chronogram (Fig. 2). All analyses were run in R v.3.6.0[72], using packages *vegan v.2.5-5, phytools v.0.6-60, geiger v.2.0.6.1, DDD v.4.0, surface v.0.5, mvMorph v.1.1.0, readxl v.1.3.1, RColorBrewer v.1.1-2, ggplot2 v.3.3.3, grid v.4.0.3, gridExtra v.2.3, dplyr v.1.0.2* and *plyr v.1.8.6*.

**Estimation of speciation dynamics**. The diversification dynamics was assessed from the lineage through time (LTT) plot on 100 randomly chosen trees[26], together with the $\gamma$-test[73]. The $\gamma$-test asks whether per-lineage speciation and extinction

rates have remained constant through time. A deviation from 0 indicates a change in speciation rates. As a null model, we simulated 1000 trees under the pure birth assumption, since the analyses under the birth-death assumption did not differ and the estimated extinction rates were negligible.

A change in diversification rate may be related to differences in rates of extinction or speciation. We searched for the best model that describes the course of diversification. To do this, we first fitted different models that account for shifts in diversification parameters: speciation rate ($\lambda$), extinction rate ($\mu$) and clade-carrying capacity ($K$)[74]. Specifically, we tested whether our data is best described by a simple birth-death model with constant $\lambda$ and $\mu$, diversity dependent models with incorporated $K$, or models with a shift in time: shift in one or more parameters $\lambda$, $K$ and $\mu$. We also estimated the time of the shift. The fit of the models was compared using Akaike weights.

**Analysis of ecological diversification**. The ecological diversification of *Niphargus* species was estimated using changes through time plots (CTT)[26]. First, discrete habitat traits were mapped onto a phylogenetic MCC tree using stochastic character mapping, assessed from 1000 simulated stochastic character histories using the tip states on the tree and a continuous-time reversible Markov model of trait evolution, fitted on the phylogeny[75]. The evolutionary rates of transition between different ecologies (i.e. transition matrix Q) were estimated from the data. We plotted the mean number of character changes per sum of branch lengths in a given time period from a set of stochastic map trees against evolutionary time. The empirical CTT was tested against the assumption of constant evolution inferred from a null model. The null Brownian-motion model was generated from 1000 simulated stochastic character maps using the tree and the observed transition matrix Q[26].

**Analysis of morphological diversification**. Morphological diversification was examined using disparity through time plots (DTT), the morphological disparity index (MDI), the ranked confidence envelope test, the node height test, and by searching for the best-fitting model of trait evolution. We used eleven continuous morphological traits (see Dataset). Morphological traits were treated as follows. First, for each trait, we calculated mean values per species. In subsequent analyses, body length was used as a raw variable, whereas other traits were corrected for the body length. We regressed all traits onto body length and calculated phylogenetically corrected residuals using phylogenetic generalised least squares[26]. DTT plots were calculated to examine the course of morphological diversification[30]. The disparity value at each node represents a variance-related estimate of the dispersion of points in multivariate space insensitive to sample size, measured as average squared Euclidean distance among all pairs of species of given subclade[30]. Disparity values of subclades were divided by the overall disparity of the clade. For each divergent event, the average of the relative disparities of all subclades whose ancestral lineages were present at that time was calculated and the empirical DTT curve was compared to null model expectations, generated from 1000 simulated trait distributions obtained from a model of evolution by Brownian motion. We tested the significance of deviation from expected disparity by two methods, pairwise and ranked confidence envelope tests (only results for more stringent, ranked confidence envelope are shown). Second, we calculated the morphological disparity index (MDI) that measures the overall difference in the relative disparity of a clade compared with that expected under the null model[31].

We also performed the univariate node height test that searches for a significant correlation between phylogenetically independent contrasts and the heights of the nodes at which they are generated[32]. The height of a node is defined as the absolute distance between the root and the most recent common ancestor of the pair from which the contrast is generated. Significant correlation indicates that the rate of trait evolution is changing systematically through the tree[32]. We performed a node height test for each of the 11 traits separately.

Finally, we explored which evolutionary model best describes trait evolution within a multi-trait framework that accounts for trait covariances[33]. To reduce the number of parameters, we calculated phylogenetically corrected PCA, and used the first two axes (cumulative variance 97.5%, Supplementary Table 3). We fitted different multivariate models of continuous trait evolution to our tree: Brownian Motion (BM), Early Burst (EB), ACcelerate-DeCelerate (ACDC), Ornstein–Uhlenbeck and models with a shift in time from one regime to another. As a shifting point, we took the estimated time of change of diversification rates (see Species diversification). The fit of the models was estimated using Akaike weights[33].

To illustrate morphological variability and parallel evolution of morphotypes we used hierarchical clustering using Ward's method, using the same dataset as for DTT. We identified nine different clusters, well segregated also when plotted onto the first two PCA axes (see above) (Supplementary Fig. 9).

**Analysis of convergence**. To test for convergence on the macroevolutionary adaptive landscape we used SURFACE (acronym for 'SURFACE Uses Regime Fitting with AIC to model Convergent Evolution'). SURFACE tests for lineages' convergence in phenotype without using a priori designations of ecomorphs[37]. Briefly, within adaptive radiation, different lineages may evolve into distinct ecological niches, each occupying its own adaptive peak. These adaptive peaks are explicitly modelled in Hansen's model of trait evolution using two parameters, trait value and strength of stabilising selection[38]. The assessment of convergence is made in two steps. In the first step, SURFACE successively fits a series of adaptive peaks onto a phylogeny using stepwise AIC. In turn, it simplifies the model by merging similar adaptive peaks using stepwise AIC. These adaptive peaks may point to convergent evolution[37]. The adaptive peaks were inferred using the first two axes of phylogenetically corrected PCA (explaining 97.5% of total variance) as a surrogate for morphological data, and the MCC phylogeny. We ran two analyses. First, we used the entire genus phylogeny. This analysis was not sensitive enough to capture evolutionary dynamics in the Pannonian and Pontic clade. These species are generally small-pore inhabitants and on average an order of magnitude smaller than in other clades. Relative morphological variation within these two clades is smaller as compared to variation among larger species and traits that vary in these clades explain only the fourth PCA axis, which was not used in the analysis. The genus-level analysis identified one and two adaptive peaks for Pontic and Pannonian clade, respectively. To account for a potential lack of sensitivity, we repeated the analysis on a pruned tree, composed of only two clades. Indeed, this analysis identified additional adaptive peaks (not reported in the main text). The results of both analyses are presented in Supplementary Fig. 5.

**Ancestral habitat and ancestral area reconstruction**. Ancestral habitat was estimated from the species habitats that were mapped onto a phylogenetic MCC tree using stochastic character mapping. We ran 1000 simulated stochastic character histories using the tip states on the tree and a continuous-time reversible Markov model of trait evolution, fitted on the phylogeny[75]. The evolutionary rates of transition between different habitats (i.e. transition matrix Q) were estimated from the data. Species with unknown habitat were given equal probability for each state. We reconstructed ancestral geographical areas for six analysed clades using the longitude and latitude of our samples mapped onto phylogeny. The evolution of ancestral ranges was mapped on the MCC tree under the Brownian motion model within a Bayesian framework implemented in Geo Model in BayesTraits V3.0.1[76]. The Geo model maps longitude and latitude onto a three-dimensional Cartesian coordinates system, and thus eliminates error in reconstructions due to the spherical nature of the Earth. We ran 11 million MCMC generations, sampled every 10,000th, discarded the first million as a burn-in, and checked the trace for convergence. The 95% confidence intervals for regions of origin were determined such that we discarded 5% of the reconstructed sites that were farthest away from the centroid. The remaining estimated points of origin were plotted onto paleo maps, adapted from the available literature[39,40]. Maps were produced using QGIS[77], and Esri World Physical Map[78].

**Reporting summary**. Further information on research design is available in the Nature Research Reporting Summary linked to this article.

## Data availability

Sequence data have been deposited in GenBank. Vouchers, GenBank accession numbers with hyperlinks to GenBank, spatial coordinates, morphometric data, and ecological data of samples are listed in Supplementary Data 1, 3 and 4. Newly obtained sequences are available in GenBank under accession numbers MT191378–MT192029, MZ270543 and MZ295224. Alignments and settings for phylogenetic analyses are available on Zenodo (https://doi.org/10.5281/zenodo.4779097)[79]. European Groundwater Crustacean Database (EGCD) used for compiling ecological information of species is accessible on http://www.freshwatermetadata.eu/metadb/bf_mdb_view.php?entryID=BF73.

## Code availability

All code used in analyses is available on Zenodo (https://doi.org/10.5281/zenodo.4779097).

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

## Acknowledgements

We are grateful to all who accompanied them during the fieldwork or contributed the samples: Florian Altermatt, Roman Alther, Dorottya Angyal, Robert Baković, Gergely Balazs, Mariella Baratti, Jana Bedek, Helena Bilandžija, Michel Blant, Sket Boris, Ivailo Borissov, Gregor Bračko, Buckle, Markus Bur, David Culver, Hrvoje Cvitanović, Dattagupta, Teo Delić, Michel Dethier, Christophe Douady, Andrej Drevenšek, Elzbieta Dumnicka, Somayeh Esmaeili, Vuilleumier Farine, Žiga Fišer, Jean-François Flot, M. Freiburghaus, Andreas Fuschs, Reinhardt Gerecke, Michal Grabowski, Christian Griebler, Dajana Hmura, Florian Hof, Paul J. Wood, Branko Jalžić, Katerina Jazbec, Lyubomir Kenderov, Alen Kirin, Lee Knight, Uroš Kunaver, Eva Lasič, Katja Lasnik, Marko Lukić, Florian Malard, Janja Matičič, Severine Matthys, M. Mede, Ioana Meleg, Giuseppe Messana, Andrej Mock, Jos Notenboom, Dmitri Palatov, Miloš Pavičević, Matija Perne, Matija Petković, B. Petrov, Slavko Polak, Pospisil, Simona Prevorčnik, Tonći Rađa, Nastassia Rajh Vilfan, Lucija Ramšak, Anja Remškar, Nathalie Salicier, Marjetka Šemrl, Damijan Šinigoj, Boris Sket, David Škufca, Edo Stloukal, Fabio Stoch, Pascal Stucki, J. Sychra, Mustafa Tanatmiş, G. Tomasin, Jernej Tramte, Tom Turk and Diether Weber. We thank Teo Delić and Denis Copilaş-Ciocianu who provided photographs of *Niphargus* specimens, used in Fig. 1. A.M. is thankful to Marjeta Konec for her help with laboratory work. Š.B., P.T., A.M. and C.F. were supported by the Slovenian Research Agency (Programme P1-0184, project J1-2464, PhD grant (Š.B., KB139 382597) and grant contract 1000-08-310028 (A.M)).

## Author contributions

C.F., P.T. and Š.B. conceived and designed the study. A.M., Š.B. and C.F. collected the data. Š.B. collected the data, conducted analyses, and prepared the results. C.F., Š.B., P.T., O.S. and A.M. wrote the manuscript.

## Competing interests

The authors declare no competing interests.
