## [Peer Review File · Nature Communications]

Reviewers' Comments:

Reviewer #1:

Remarks to the Author:

Review of the manuscript

"The Atlantis of Europe's adaptive radiations"

by Špela Borko, Peter Trontelj, Ole Seehausen, Ajda Moškrič, Cene Fišer

The manuscript presented to me for reviewing is a very interesting contribution, dealing with an entirely subterranean adaptive radiation of the amphipod's *Niphargus* in Europe. The authors approached the problem by reconstructing a time-calibrated phylogeny. The LTT, CTT and DTT analyses indicated that uplift of carbonate massifs provided ecological opportunities, promoting adaptive radiation. The subterranean system shaded from the climatic fluctuations, and it reserved the ancient adaptive radiations. All the analytical methods are appropriate for such kind of datasets. Thus, the obtained results are reliable and well support the drawn conclusions. Apart of that, there are several points listed below that need to be addressed by the authors prior to acceptance of the manuscript for publication:

1) Title: "Atlantis" is uneasy to understand and does not appear in the text. The meaning should be explained in somewhere.

2) Abstract: The authors performed many analyses on morphological and ecological traits; however, there are no words on it in the abstract.

3) Introduction, line 31: The sentence "bares little evidence of extensive ARs as known from other continents" should be "bears".

4) Results, line 89 and Discussion line 172: "first 20-30 Mya" or "first 20-30 My" should be 20-30 million years.

5) Results, line 108: "most morphological variation of the entire genus is partitioned among-clades rather than within clades." But from supplementary Figure 5 the morphotypes can not be distinguished clearly. Can you please discuss this a bit more?

6) Results, line 116: "diversification (Fig. 3C)" should be "diversification (Fig. 3A)"

7) The n-dash and m-dash should be distinguished throughout the text including the references.

8) Fig.1: The sentence "Some morphotypes inhabit the same type of habitat (e.g., cave lake, lake giant and daddy-longleg in deep stagnant waters), and conversely, distinct habitats can be inhabited by species with similar morphology (shallow subterranean and cave stream, fissure 664 system and slender interstitial)" should be rephrased. It is rather obscure, meaning no relationship between ecological habitat and morphotypes?

9) Fig.3c: Fig.3c is constructed using 11 morphological characters or not? Therefore, Fig.3b and Fig.3c are different topology. Please explain clearly, how to connect Fig.3b and Fig.3c? I suggest use the same tree to describe the habitat and morphology like Fig.4. Legends Fig. 3 "grey and black arrows on figures b and c point to species from a cave (Vjetrenica Cave System, Bosnia and Hercegovina) and interstitial", but in Fig. 3, grey arrows for interstitial and black arrows for caves.

Concluding, I assume the manuscript to be worth publishing in the Nature Communications after corrections/comments required in the points listed above, are addressed by the authors.

Reviewer #2:

Remarks to the Author:

In this paper the authors have used molecular phylogenetics of many *Niphargus* samples to infer the tempo and distribution ecological of speciation in this group. They find a burst of evolutionary diversification also associated with diversification into an ecological niche, which is typical of an adaptive radiation. They also assess the of overlap extent across different clades to infer signatures of parallelism in radiations. They reconstruct the origins of the different clades in paleoenvironments. This is a largely understudied group with a deep history Europe, which adds to the novelty. Phylogenetic analyses are fairly standard and seem to be appropriately conducted.

I have, however, some constructive criticisms.

1. The sequencing dataset is over-sold with possible consequences for rate-based analyses. In fact, from table S3 and methods it is evident that there are many taxa for which only one gene is used. Some missing data is expected and not predicted to cause issues, but this is quite a lot of heterogeneity. Was there any exploration of the role of gene in the or missing data in the analysis? This is especially relevant for calculations of change through time. It would be appropriate to show the influence of different markers on the reconstruction.
2. The morphological analyses and results are not sufficiently presented to allow meaningful interpretation of the disparity
3. All dates given in the ms should be presented with HPDs or as ranges. A lot of interpretation is based on this statement “.. corresponds to emerging karstic regions in South-Eastern Europe at about 15 Mya” but more specific time period evidence and detailed background is needed. At many points the dates seem pseudo-precise That said, the single radiation point is evident in the tree.
4. The analysis in Fig 4 is not convincing evidence of parallelism, and is not a new analysis as it should all be already included in the genus-wide analysis. I also do not see how this section answers the topic of “Independent parallel radiations” as, from what I understand, there is no parallelism or convergence tested. This part (and associated discussion) needs more robust analysis to address the question posed and a re-evaluation.

Minor

1. Table 2 is could be made more comprehensible
2. “Testing alternative models of diversification, we identified an increase in diversification between 15.49 and 15.70 Mya” this is surely pseudoprecise
3. Where is the evidence for this statement? “Morphological Disparity Index²⁷ that measures the overall difference in relative disparity of a clade against the disparity expected under the null model, showed that most morphological variation of the entire genus is partitioned among-clades rather than within clades (MDI = 0.028, ranked envelope test p interval 0.0009 – 0.0270).”
4. Is “disparafication” a word? I don’t think it’s helpful to develop this (new?) term here.
5. What is with the chimeras? (table S2) This is not covered at all.
6. There are some minor grammatical errors to be corrected
7. This statement needs a citation “the largest phylogeny of any subterranean clade so far”.

Point-by-point response to reviewers

Reviewers' comments are copied in *blue italic text*. Under each comment, a detailed response follows, with citations of relevant sections of the manuscript.

REVIEWER #1 (REMARKS TO THE AUTHOR):

The manuscript presented to me for reviewing is a very interesting contribution, dealing with an entirely subterranean adaptive radiation of the amphipod's Niphargus in Europe. The authors approached the problem by reconstructing a time-calibrated phylogeny. The LTT, CTT and DTT analyses indicated that uplift of carbonate massifs provided ecological opportunities, promoting adaptive radiation. The subterranean system shaded from the climatic fluctuations, and it reserved the ancient adaptive radiations. All the analytical methods are appropriate for such kind of datasets. Thus, the obtained results are reliable and well support the drawn conclusions. Apart of that, there are several points listed below that need to be addressed by the authors prior to acceptance of the manuscript for publication:

1) Title: "Atlantis" is uneasy to understand and does not appear in the text. The meaning should be explained in somewhere.

Thank you for pointing this out. We included explanatory statement in the Introduction (lines 60–61):
"This mostly hidden and isolated ecological space allegorically resembles 'Atlantis', the sunken mythological civilisation."

2) Abstract: The authors performed many analyses on morphological and ecological traits; however, there are no words on it in the abstract.

We changed "Analyses of morphological and ecological traits using a time-calibrated multilocus phylogeny..." into more concise "Modelling lineage diversification and evolution of morphological and ecological traits using a time-calibrated multilocus phylogeny..." (lines 20–21).

3) Introduction, line 31: The sentence "bares little evidence of extensive ARs as known from other continents" should be "bears".

It should stay *bears* as to *uncover*.

4) Results, line 89 and Discussion line 172: "first 20-30 Mya" or "first 20-30 My" should be 20–30 million years.

We corrected My in million years in both cases.

5) Results, line 108: "most morphological variation of the entire genus is partitioned among-clades rather than within clades." But from supplementary Figure 5 the morphotypes can not be distinguished clearly. Can you please discuss this a bit more?

We are unsure if we understand the question correctly. We tried to address it as broad as possible but we are looking forward to additional elaboration of the reviewer's concern.

Firstly, we clarified the Supplementary Figure 5 (in new version of MS numbered as Supplementary Figure 9) with adding a caption to it, that states:

"Plot of Phylogenetically corrected PCA, inferred from raw values of eleven morphological traits. Values are coloured by clusters, obtained from hierarchical Ward's clustering of same traits, regressed onto body length (see Fig. 2C). First two axes cover 97.5% of variability and are explained by body, antennae and leg length. Clusters are well reflected also on PCA plot, although some overlap is present. Note that clusters that are least distinguished by PCA are Small pore stout, short appendages and Small pore slender, short appendages, that partially also overlap with Undistinct cluster. Slenderness or stoutness is represented by depths of coxal plates

and pereopod bases, which explain the fourth PCA axis; and undistinct morphotype is a generalist one, that falls in between other categories and can be found in a variety of habitats.”

Second, we would like to state that our dataset is gathered to represent the whole variety of morphological continuum of *Niphargus*, including undistinctive generalist species. Regardless of the continuous morphospace, distinct morphotypes can be described and explained by properties of the ecological niche they occupy. The morphotypes are most clearly expressed in species of different morphotypes that live in syntopies in distinct subterranean habitats (up to nine species per site, see Figure 2).

Third, we acknowledge a poorly phrased section of the first manuscript that refers to DTT analysis. We rewrote this part, clarified the Results and Methods sections and explained those changes in detail in response to Q2 and Q4 of Reviewer2 (see below).

6) Results, line 116: “diversification (Fig. 3C)” should be “diversification (Fig. 3A)”

We corrected the sentence.

7) The n-dash and m-dash should be distinguished throughout the text including the references.

We corrected the dashes in the text and also in the references.

8) Fig.1: The sentence “Some morphotypes inhabit the same type of habitat (e.g., cave lake, lake giant and daddy-longleg in deep stagnant waters), and conversely, distinct habitats can be inhabited by species with similar morphology (shallow subterranean and cave stream, fissure 664 system and slender interstitial)” should be rephrased. It is rather obscure, meaning no relationship between ecological habitat and morphotypes?

Thank you for pointing out that caption is not clear enough. We included additional explanation into the Results section, to clarify complex relationship between different aspects of *Niphargus* ecology (lines 107–113). Specifically, we stated:

“There is no simple one-to-one correspondence between the morphotypes (Fig. 1) and habitats listed above. Different morphotypes sometimes share the same habitat but may differ in their trophic niches. Such examples are species of different body size co-inhabiting cave lakes¹⁷, or species of different body shapes coexisting in interstitial groundwater²⁰ (see Fig 2b for an example). Conversely, some species with generalist morphology can be found in more than one subterranean habitat²⁷.”

Second, we corrected the Figure 1 in a way, that the descriptions on the figure now state habitat of each species (all caps) and descriptive morphotype (small caps). We also corrected the caption:

“Morphological and ecological diversity of *Niphargus* species. Adaptive radiation of *Niphargus* produced several morphotypes that inhabit distinct subterranean habitats and niches. Different-sized morphotypes can occur together in cave lakes, and differently shaped morphotypes can occur together in interstitial habitats¹⁷. These morphotypes evolved in at least five adaptive radiation events from hypothetical small-bodied ancestors in the shallow interstitial^{20,40}. The habitat and the description of morphotype is given along each species.”

Third, see also a chapter Morphotype description in Supplementary Methods, that states:

“Morphometric traits roughly define several morphological types (Fig. 1). The relationship between morphology and habitat occupancy is imperfect, but relatively well expressed in synoptic populations²⁴. Species (<10 mm) that live in fissure system (habitat category 1) are small with appendages of variable length; interstitial species (habitat category 2) can be divided into slender-long-legged and stout-short-legged ones (interstitial slender and interstitial stout in Fig. 1). Phreatic lakes (habitat category 3) harbour three morphotypes, all long legged: cave lake (large 15-23 mm), lake giants (>25 mm) and daddy-longlegs (large >20 mm, appendages longer than body length). Stream morphotypes, from two size classes, moderately large (10-15 mm) to large (15-25 mm) with short appendages, inhabit cave streams and shallow subterranean (habitat

categories 4 and 5) and cannot be told apart by morphology. Chemical properties of habitat have no effect on morphology. From chemically distinct habitats were found slender interstitial, cave and shallow subterranean stream, lake and lake giant ecomorphs.”

9) Fig.3c: Fig.3c is constructed using 11 morphological characters or not? Therefore, Fig.3b and Fig.3c are different topology. Please explain clearly, how to connect Fig.3b and Fig.3c? I suggest use the same tree to describe the habitat and morphology like Fig.4.

Legends Fig. 3 “grey and black arrows on figures b and c point to species from a cave (Vjetrenica Cave System, Bosnia and Hercegovina) and interstitial”, but in Fig. 3, grey arrows for interstitial and black arrows for caves.

We thank for pointing out unclear relationship between three aspects of evolution of *Niphargus*. We corrected the figure (in new version of MS numbered as Figure 2) in and its caption to more clearly state that it shows: a) Through Time plots; b) calibrated phylogeny with marked habitats; c) cluster dendrogram of 11 morphological traits. We hope it is clear now that b) and c) are not the same phylogeny, coloured different ways but two different analyses.

We corrected the caption of the Figure 3:

“Three different aspects of the adaptive radiation of *Niphargus*. a, Lineage diversification, ecological and morphological disparification notably accelerated approximately 15 Mya. The pattern is well visible on LTT, CTT and DTT plots. The time of evolutionary shift is indicated by red arrows and grey line. b, Calibrated phylogeny, species’ habitats and ancestral habitat reconstructions of *Niphargus*. Tips are labelled according to extant habitat. Pies on selected nodes represent reconstructed ancestral habitat (for complete reconstructions see Supplementary Fig. 2). Clades that were subjected to further analyses are coloured and named. c, Cluster dendrogram based on eleven functional morphological traits. The same groups were partially recovered by PCA (Supp. Fig. 8). Phylogenetic composition of morphological groups is labelled on dendrogram. Cluster analysis shows that unrelated species evolved similar morphology (see clade’s acronyms on basal nodes of dendrogram). High morphological disparity presumably allowed high levels of syntopy and the formation of species rich communities of closely related species. As an example, black and grey arrows on figures b and c point to species from a cave (Vjetrenica Cave System, Bosnia and Hercegovina) and interstitial (Sava river close to Medvode, Slovenia) communities, respectively. Note that many community members are closely related (same clade) but belong to different morphological clusters.”

Second, we thank for pointing out the erroneously coloured arrows – we corrected the mistake in a way that black represents cave and grey represents interstitial on both subfigures.

REVIEWER #2 (REMARKS TO THE AUTHOR):

In this paper the authors have used molecular phylogenetics of many *Niphargus* samples to infer the tempo and distribution ecological of speciation in this group. They find a burst of evolutionary diversification also associated with diversification into an ecological niche, which is typical of an adaptive radiation. They also assess the of overlap extent across different clades to infer signatures of parallelism in radiations. They reconstruct the origins of the different clades in paleoenvironments. This is a largely understudied group with a deep history Europe, which adds to the novelty. Phylogenetic analyses are fairly standard and seem to be appropriately conducted. I have, however, some constructive criticisms.

1. The sequencing dataset is over-sold with possible consequences for rate-based analyses. In fact, from table S3 and methods it is evident that there are many taxa for which only one gene is used. Some missing data is expected and not predicted to cause issues, but this is quite a lot of heterogeneity. Was there any

exploration of the role of gene in the or missing data in the analysis? This is especially relevant for calculations of change through time. It would be appropriate to show the influence of different markers on the reconstruction.

We agree with the reviewer and thank for constructive comment. Robustness with respect to missing genes was our concern from the start. We have conducted analyses on datasets of different size with different degree of missing sequences, however the results were not included in the first submission. We included additional results and clarifications into the revised manuscript and below:

The first main source of missing data are single marker MOTUs, with only COI sequence available. To assess the potential influence of missing sequence data in the data set, we repeated the BEAST 2 and through time analyses (LTT, CTT, DTT and subsequent statistics) on a subset of 301 *Niphargus* MOTUs with at least two markers (Supplementary Figure 9a, b). The results are congruent with the results of extended data set and show negligible effect of missing data onto final phylogeny reconstruction and subsequent analyses. The remarkable change in tempo and mode of diversification of *Niphargus* is retained in the results based on abovementioned subset and points towards strong pattern, present in the data. We incorporated this issue into the main text, in Methods (lines 380–385):

“To assess the potential influence of missing sequence data in the data set, we repeated the BEAST 2 and through time analyses (LTT, CTT, DTT and subsequent statistics) on a subset of 301 *Niphargus* MOTUs with at least two markers (Supplementary Fig. 8). The results were congruent with the results obtained from the extended data set and showed negligible effect of missing data on the final phylogeny reconstruction and downstream analyses.”

The second important source of missing data in alignment is 18S marker. 18S marker is longest marker used (2109 bp). It is a highly conserved marker of slow-evolving characters used to obtain evolutionary information on deep nodes. We sequenced it only from a subset of representatives of distinct clades (113 MOTUs), recognised already in previous studies.

2. The morphological analyses and results are not sufficiently presented to allow meaningful interpretation of the disparity

We thank the reviewer for the comment. It is unclear what exactly the reviewer had in mind, but we rephrased and extended the sections that refer to disparity in Results and Methods. However, we kindly ask the reviewer to elaborate the question if there are still unanswered concerns.

First, we added additional analysis of rate evolution (node height test), that is complementary to already provided MDI and rank envelope test (Supplementary Table 1 and Supplementary Figure 3). We clarified this section in Results: we expanded the descriptions of aims, results and interpretation of each method (lines 114–144):

“In the next step, we explored the evolutionary dynamics of morphological disparity within *Niphargus*, using disparity through time plots (DTT)²⁸. Eleven morphological quantitative traits served as proxy for the ecological function of species^{17,20} (Supplementary Data 4). The DTT approach enables the investigation of disparity patterns in conjunction with clade age. For each divergent event (i.e., each node) we calculated the average of the disparities of all subclades whose ancestral lineages were present at that time, standardized against the overall disparity. Disparity values near 0 imply that subclades contain relatively little of the variation present within the entire clade, and most variation is partitioned as differences between subclades. Values near 1 imply that subclades contain a substantial proportion of the total variation, indicating that species within subclades have independently evolved to occupy similar regions of the ecomorphological space^{28,29}. The observed DTT curve was compared to the null hypothesis of neutral evolution of morphology in which we simulated the relative disparities obtained from Brownian motion model²⁸. The DTT plot suggested an early burst of disparity when the two major lineages arose approximately 35 million years ago, followed by 15 million years of continuous disparity decline that is in accord with neutral models of evolution. This decline is

sharply ended by a significant recovery of morphological disparity at 15 Mya, pointing towards independent diversification of ecological traits within subclades. This phase of phenotypic diversification coincides with increase in the rates of species accumulation and their ecological diversification (Fig. 2: LTT and CTT). The morphological disparity index (MDI)²⁹ that measures the overall difference in relative disparity of a clade against the disparity expected under the null model, showed insignificantly higher disparity than expected by the null model (MDI = 0.028, $p = 0.4$). We attribute this insignificant result to the overall DTT dynamics where low disparities in early history cancelled out higher values during the last 15 Mya. The rank envelope test that tests how extreme the reconstructed DTT curve is compared to the simulated curves, showed that the DTT curve falls outside the expected 95% confidence intervals of the null model simulations (p interval 0.0009–0.0270). Visual inspection of the DTT plot (Fig. 2) showed that this deviation happened from 15 Mya onward. We also tested whether the evolutionary rates of the eleven studied traits changed in time using the univariate node height test^{29,30}. The results were significant for all eleven traits, showing that the rate of their evolution indeed systematically increased during the evolutionary history of the genus (Supplementary Table 1, Supplementary Fig. 3).”

We appropriately changed the Methods section as well (lines 426–451):

“Morphological diversification was examined using disparity through time plots (DTT), the morphological disparity index (MDI), the ranked confidence envelope test, the node height test, and by searching for the best-fitting model of trait evolution. We used eleven continuous morphological traits (see Dataset). Morphological traits were treated as follows. First, for each trait we calculated mean values per species. In subsequent analyses, body length was used as raw variable, whereas other traits were corrected for the body length. We regressed all traits onto body length and calculated phylogenetically corrected residuals using phylogenetic generalized least squares²⁴. DTT plots were calculated to examine the course of morphological diversification²⁸. The disparity value at each node represents a variance-related estimate of the dispersion of points in multivariate space insensitive to sample size, measured as average squared Euclidean distance among all pairs of species of given subclade²⁸. Disparity values of subclades were divided by the overall disparity of the clade. For each divergent event, the average of the relative disparities of all subclades whose ancestral lineages were present at that time was calculated and the empirical DTT curve was compared to null model expectations, generated from 1000 simulated trait distributions obtained from a model of evolution by Brownian motion. We tested the significance of deviation from expected disparity by two methods, pairwise and ranked confidence envelope tests (only results for more stringent, ranked confidence envelope are shown). Second, we calculated the morphological disparity index (MDI) that measures the overall difference in relative disparity of a clade compared with that expected under the null model²⁹. We also performed the univariate node height test that searches for significant correlation between phylogenetically independent contrasts and the heights of the nodes at which they are generated³⁰. The height of a node is defined as the absolute distance between the root and the most recent common ancestor of the pair from which the contrast is generated. Significant correlation indicates that the rate of trait evolution is changing systematically through the tree³⁰. We performed node height test for each of 11 traits separately.”

3. All dates given in the ms should be presented with HPDs or as ranges. A lot of interpretation is based on this statement “.. corresponds to emerging karstic regions in South-Eastern Europe at about 15 Mya” but more specific time period evidence and detailed background is needed. At many points the dates seem pseudo-precise That said, the single radiation point is evident in the tree.

We addressed all aspects of this comment.

In Results, we included detailed background for paleogeographic and geological history of Europe, together with uncertainty in the dating, as ranges of timing of different events (lines 195–208):

“The increase in diversification and disparification around 15 Mya corresponds to the emergence of karstic regions in South-Eastern Europe in the area of the modern South-Eastern Alps, the Dinarides and the

Carpathians^{35,36}. This area underwent a complex geological history. Collision of the European and Adriatic slabs during the Eocene (66–34 Mya) caused vivid tectonic movements that resulted in drastic topographic changes³⁷. In the Oligocene (32–23 Mya), the South-Eastern Alps and the Dinarides emerged from the surrounding seas, and the uplift of the Carpathians begun. The exposure of carbonate rocks of the Alpine-Carpathian-Dinaridic orogenic system to atmospheric processes triggered karstification, i.e., the formation of caves and a variety of other subterranean habitats^{17,25}. This process begun in the Oligocene and peaked in the Early Miocene (23–16 Mya). In that period, mountain uplift continued and the three mountain ranges acted as islands in the Paratethys Sea, occasionally connected during marine regressions. During the later Miocene (16–14 Mya), a mosaic of saline and freshwater aquatic environments and new emerging carbonate islands replaced the Paratethys, which completely regressed from Late Miocene onwards (11 Mya)^{38,39}.”

We included HPDs wherever we discuss divergence events as reconstructed with time-calibrated phylogeny and stated explicitly that HPDs are given in supplementary figures.

Specifically in lines 75–77:

“The genus *Niphargus* originated in the Middle Eocene (mean value 47 Mya, highest posterior density (HPD) interval 39–56 Mya) in the region that nowadays represents Western Europe (Fig. 3, Supplementary Data 2, Supplementary Fig. 1).”

and lines 165–168:

“The Pontic and the Pannonian clade diverged from the rest of *Niphargus* early (38 Mya, HPD 40–35 Mya), and split 29 Mya (HPD 34–24 Mya). The South Dinaric, West Balkan, North Dinaric and the Apennine clade are younger, they started to diversify at 15–11 Mya (Supplementary Fig. 1).”

We also corrected later mentions of timing of geological events as intervals.

4. The analysis in Fig 4 is not convincing evidence of parallelism, and is not a new analysis as it should all be already included in the genus-wide analysis. I also do not see how this section answers the topic of “Independent parallel radiations” as, from what I understand, there is no parallelism or convergence tested. This part (and associated discussion) needs more robust analysis to address the question posed and a re-evaluation.

We thank the reviewer to point out the question of parallelism. We addressed this issue in multiple ways.

1. We acknowledge that we did not explicitly test for parallelism, so we changed the manuscript accordingly, removing the implication of parallel radiations through the text and also in the subtitle of the Results:

“Multiple independent radiations.”

2. We would like to state that we refer to six clades as examples of independent radiations. These clades are not exact replicates, but can be seen as individual ARs, in slightly different settings, that lead to similar, yet different outcomes, as we state from lines 169–178:

“The analyses of diversification (LTT plots and γ -test) suggested that all clades display the pattern of an early burst of species accumulation with onset between 15 and 5 Mya. CTT plots did not deviate from null models, but DTT analyses imply adaptive radiation patterns in the Pontic, Pannonian, West Balkan and the North Dinaric radiations (Table 4, Fig. 4). Dynamics of species accumulation and ecological disparification among these four clades differ (Table 4, Fig. 4), possibly reflecting differences in habitat diversity among regions where these radiations unfolded or differences in the time of arrival of these lineages in the novel habitats. The Apennine clade is composed mainly of morphologically similar, still undescribed species. Because of lack of morphological data, we could not derive conclusions about the nature of this radiation.”

3. Importantly, these clades are phylogenetically and partially geographically independent, as we explained in detail in lines 161–168:

“We selected six well-supported reciprocally monophyletic clades that were geographically well-defined and sufficiently large ($N \geq 25$ species) to be explored for patterns of AR by repeating LTT, CTT and DTT analyses on each clade separately. They are distributed mostly in the karstic regions of Italy and South-Eastern Europe and spatially overlap to various extent (Table 3, Fig. 2, Supplementary Fig. 4). The Pontic and the Pannonian clade diverged from the rest of *Niphargus* early (38 Mya, HPD 40–35 Mya), and split 29 Mya (HPD 34–24 Mya). The South Dinaric, West Balkan, North Dinaric and the Apennine clade are younger, they started to diversify at 15–11 Mya (Supplementary Fig. 1).”

4. We additionally addressed geographical distribution in Supplementary Figure 4, where we show the distribution of each analysed clade, as species occurrence data from our up-to-date database (SubBio DB). The detailed analysis of distributions (not shown) shows that clade occurrences are mutually exclusive in many cases. This is an ongoing study and not all occurrence data is published yet, so we kindly ask you to allow the omission of publishing any additional information.

5. From the reconstruction of ancestral states using stochastic habitat mapping (Figure 2, Figure 3, Supplementary Fig. 2) it is evident that “colonization and adaptation to new subterranean habitats that took place repeatedly in different clades” as stated in lines 92–93.

6. Although the detailed analysis of convergence is out of scope of this study, we nevertheless performed preliminary test of convergent evolution, as presented in the Results lines 179–193. Specifically we performed the SURFACE analysis:

“Finally, we explored whether these clade-level radiations show signs of between- or within clade convergent evolution. We used SURFACE, a method that tests whether lineages have converged in phenotype without using a priori designations of ecomorphs³³. It applies Hansen’s model of adaptive peaks³⁴ modelled onto the phylogeny and assumes that in the case of convergence similar phenotypes in different clades could be attributed to the same adaptive peaks. Three models that best explained the evolution of ecomorphological traits predicted 14 to 16 different adaptive peaks, of which 11 to 12 were found to be convergent whereas only three to four were unique. These peaks partially correspond to clusters from the morphological analysis (Fig. 2). Of 11 convergent peaks, two evolved multiple times within one clade, whereas nine peaks evolved in two or more focal or non-focal clades. The results of the best model are illustrated in Supplementary Fig. 5. We tentatively conclude that at least four out of six large and phylogenetically distinct clades could be considered as adaptive radiations. Although showing some degree of convergence, the radiations overall adapted to distinct sets of adaptive optima, especially so among the South-Eastern Europe clades.”

We included the methodology in Methods section, lines 464–485:

“To test for convergence on the macroevolutionary adaptive landscape we used SURFACE (acronym for “SURFACE Uses Regime Fitting with AIC to model Convergent Evolution”). SURFACE tests for lineages’ convergence in phenotype without using a priori designations of ecomorphs³³. Briefly, within adaptive radiation, different lineages may evolve into distinct ecological niches, each occupying its own adaptive peak. These adaptive peaks are explicitly modelled in Hansen’s model of trait evolution using two parameters, trait value and strength of stabilizing selection³⁴. The assessment of convergence is made in two steps. In the first step, SURFACE successively fits a series of adaptive peaks onto a phylogeny using stepwise AIC. In turn, it simplifies the model by merging similar adaptive peaks using stepwise AIC. These adaptive peaks may point to convergent evolution³³. The adaptive peaks were inferred using the first two axes of phylogenetically corrected PCA (explaining 97.5% of total variance) as surrogate for morphological data, and the MCC phylogeny. We ran two analyses. First, we used the entire genus phylogeny. This analysis was not sensitive enough to capture evolutionary dynamics in Pannonian and Pontic clade. These species are generally small-pore inhabitants and on average an order of magnitude smaller than in other clades. Relative morphological variation within these

two clades is smaller as compared to variation among larger species and traits that vary in these clades explain only the fourth PCA axis, which was not used in the analysis. Genus-level analysis identified one and two adaptive peaks for Pontic and Pannonian clade, respectively. To account for a potentially lack of sensitivity, we repeated the analysis on a pruned tree, composed of only two clades. Indeed, this analysis identified additional adaptive peaks, not reported in the main text. The results of both analyses are presented Supplementary Fig. 5.”

7. We want to state that we repeated through-time analyses again on each clade separately. In this way we showed that the radiation pattern, evident from the analysis of the whole genus is repeating on finer scale, with important differences among clades, possibly reflecting differences in habitat diversity among regions where these radiations unfolded.

8. Finally, we stated in the Results that although showing some degree of convergence, the radiations overall adapted to distinct sets of adaptive optima (lines 190–193):

“We tentatively conclude that at least five out of six large and phylogenetically distinct clades could be considered as adaptive radiations. Although showing some degree of convergence, the radiations overall adapted to distinct sets of adaptive optima ...”

We hypothesise that this pattern may be due to early habitat diversification, apparent in ancestral habitat reconstructions (lines 244–247):

“Early habitat diversification, detected in tree-wide CTT analysis but not at the level of an individual clade, may have constrained further clade-level morphological diversification. Subsequent morphological diversification of clades predominantly unfolded within one or few habitat types ...”

Minor

1. Table 2 is could be made more comprehensible

We removed columns, unnecessary to interpret the results of modelling: log likelihood (we report AIC values), degrees of freedom, convergence status (reported in caption), AIC difference (can be inferred from AIC). We also relabelled model descriptions into more readable format (first column of Tables 1 and 2).

2. “Testing alternative models of diversification, we identified an increase in diversification between 15.49 and 15.70 Mya” this is surely pseudoprecise

We change the statement to clarify, that these are results of two best models, and not an interval. The method does not provide the estimates of discussed shift parameter (lines 80–83). We also omitted decimals through all the text.

“Testing alternative models of diversification, we identified an increase in diversification between 15 and 16 Mya corresponding to increased speciation (best model), or increased speciation together with increased carrying capacity (suboptimal model), rather than decreased extinction rates.”

3. Where is the evidence for this statement? “Morphological Disparity Index²⁷ that measures the overall difference in relative disparity of a clade against the disparity expected under the null model, showed that most morphological variation of the entire genus is partitioned among-clades rather than within clades (MDI = 0.028, ranked envelope test p interval 0.0009 – 0.0270).”

We are thankful for this comment, as it pointed out an error in the description of the method. We corrected and expanded this section and included explanations and clarifications of the interpretation of morphological disparity. Changes are already presented in answer to Q2, but DDT and MDI related sections are copied here again for easier tracking (lines 114–141):

“In the next step, we explored the evolutionary dynamics of morphological disparity within *Niphargus*, using disparity through time plots (DTT)²⁸. Eleven morphological quantitative traits served as proxy for the ecological function of species^{17,20} (Supplementary Data 4). The DTT approach enables the investigation of disparity

patterns in conjunction with clade age. For each divergent event (i.e., each node) we calculated the average of the disparities of all subclades whose ancestral lineages were present at that time, standardized against the overall disparity. Disparity values near 0 imply that subclades contain relatively little of the variation present within the entire clade, and most variation is partitioned as differences between subclades. Values near 1 imply that subclades contain a substantial proportion of the total variation, indicating that species within subclades have independently evolved to occupy similar regions of the ecomorphological space^{28,29}. The observed DTT curve was compared to the null hypothesis of neutral evolution of morphology in which we simulated the relative disparities obtained from Brownian motion model²⁸. The DTT plot suggested an early burst of disparity when the two major lineages arose approximately 35 million years ago, followed by 15 million years of continuous disparity decline that is in accord with neutral models of evolution. This decline is sharply ended by a significant recovery of morphological disparity at 15 Mya, pointing towards independent diversification of ecological traits within subclades. This phase of phenotypic diversification coincides with increase in the rates of species accumulation and their ecological diversification (Fig. 2: LTT and CTT). The morphological disparity index (MDI)²⁹ that measures the overall difference in relative disparity of a clade against the disparity expected under the null model, showed insignificantly higher disparity than expected by the null model (MDI = 0.028, $p = 0.4$). We attribute this insignificant result to the overall DTT dynamics where low disparities in early history cancelled out higher values during the last 15 Mya. The rank envelope test that tests how extreme the reconstructed DTT curve is compared to the simulated curves, showed that the DTT curve falls outside the expected 95% confidence intervals of the null model simulations (p interval 0.0009–0.0270). Visual inspection of the DTT plot (Fig. 2) showed that this deviation happened from 15 Mya onward.”

We appropriately changed the Methods section as well (line 426–445):

“Morphological diversification was examined using disparity through time plots (DTT), the morphological disparity index (MDI), the ranked confidence envelope test, the node height test, and by searching for the best-fitting model of trait evolution. We used eleven continuous morphological traits (see Dataset). Morphological traits were treated as follows. First, for each trait we calculated mean values per species. In subsequent analyses, body length was used as raw variable, whereas other traits were corrected for the body length. We regressed all traits onto body length and calculated phylogenetically corrected residuals using phylogenetic generalized least squares²⁴. DTT plots were calculated to examine the course of morphological diversification²⁸. The disparity value at each node represents a variance-related estimate of the dispersion of points in multivariate space insensitive to sample size, measured as average squared Euclidean distance among all pairs of species of given subclade²⁸. Disparity values of subclades were divided by the overall disparity of the clade. For each divergent event, the average of the relative disparities of all subclades whose ancestral lineages were present at that time was calculated and the empirical DTT curve was compared to null model expectations, generated from 1000 simulated trait distributions obtained from a model of evolution by Brownian motion. We tested the significance of deviation from expected disparity by two methods, pairwise and ranked confidence envelope tests (only results for more stringent, ranked confidence envelope are shown). Second, we calculated the morphological disparity index (MDI) that measures the overall difference in relative disparity of a clade compared with that expected under the null model²⁹.”

4. Is “disparafication” a word? I don’t think it’s helpful to develop this (new?) term here.

Thank you for pointing out this error. We meant *disparification*, as for example in Baldwin 2019 (<https://doi.org/10.1111/nph.15961>). We corrected the wrong pronunciation (*disparafication*) in Figure 3 (Figure 2 in new version of MS) caption.

5. What is with the chimeras? (table S2) This is not covered at all.

Thank you for pointing this out. We inserted the explanation in the methods section (lines 322–327): “Markers that were sequenced from several specimens of the same MOTU are marked as chimeras in Supplementary Data 1. To exclude misidentification, chimeric specimens were from the same or nearby

localities and with identical sequences of overlapping markers of high resolution (e.g. COI). In the dataset, 301 MOTUs were represented by at least two fragments. For 76 MOTUs only the COI fragment was available.”

6. *There are some minor grammatical errors to be corrected*

We carefully read the manuscript and corrected grammatical errors.

7. *This statement needs a citation “the largest phylogeny of any subterranean clade so far”.*

We included reference of second large phylogeny of mostly subterranean clade, that is nevertheless twice smaller in number of MOTUs and not all lineages are subterranean.

“We reconstructed the most complete multilocus phylogeny of *Niphargus*²² (Fig. 2), containing more than twice as many species as the next largest subterranean clade analysed so far²³.”

23. Morvan, C. et al. Timetree of Aselloidea Reveals Species Diversification Dynamics in Groundwater. *Syst. Biol.* 62, 4, 512–522 (2013).

Reviewers' Comments:

Reviewer #1:

Remarks to the Author:

Comments to the Author

As I stated in my first review, I very much like the topic. This submission is a great improvement by the Authors in the overall writing. They include much of the discussion points I suggested. I have some minor comments before it is ready for publication.

Minor Comments

1. Fig.2, Fig4, LTT plot "log(lineages) should be ln (lineages)?" please confirm it.

I look forward to seeing this review in print.

Reviewer #2:

Remarks to the Author:

This is a re-review and so I will keep my comments focused on the changes of the revised version. I think the authors have appropriately revised the ms from the previous round of comments.

Only one remaining part from previous -

L72 -I still think more openness is needed here about the data matrix is needed. I'd suggest 'up to 7076 bp' for example and listing the mean and variance on number of bp per MOTU.

I appreciate that the authors have increased transparency on this in the supplementary and repeated the analysis with the full matrix samples.

I have some new/minor comments on presentation

I sort of agree with the other reviewer that it should be "bears little evidence". One could not substitute "uncovers" for bares in the sentence as written and have the grammar hold. But regardless the sentence is not good. At least it would need "The part of the world whose natural history has been most thoroughly explored has shown little evidence of the extensive ARs as found in other continents" However scientifically, I don't think this is a good sentence and it is not cited or particularly well supported. Contemporary Europe is small and temperate, and not even discrete so cannot be fairly compared to (m)any other continents. I'd suggest something like "This part of the world, whose natural history has been most thoroughly explored, has shown little evidence of ARs".

Line 46-47 is not a sentence. End with "is high"?

End of line 53 should have citations of examples, I think

L 63 - it's not clear to me if the sentence following from here is the predictions that are being tested or if the first sentence of the ms is the predictions being tested. I think the former, but being very clear at this important point is necessary. Maybe start with "Specifically, we test for ..."

The transition between data, background and result is not very clear. Maybe L 75 should start "We found that the genus ..."

L76 I'd suggest to revise to "in the region that is contemporary Western Europe"

L154 – the emergence of new habitats at 15 mya, independent of the findings here, was not very evident to the reader among the geological history given so far. Please refer to the figures and citation(s) needed.

No comma in line 281

some of the date ranges are flipping around from oldest to youngest and to youngest to oldest eg in the HPDs

Point-by-point response to reviewers

Reviewers' comments are copied in **blue italic text**. Under each comment, a detailed response follows, with citations of relevant sections of the manuscript.

Reviewer #1 (Remarks to the Author):

Comments to the Author

As I stated in my first review, I very much like the topic. This submission is a great improvement by the Authors in the overall writing. They include much of the discussion points I suggested. I have some minor comments before it is ready for publication.

Minor Comments

1. Fig.2, Fig4, LTT plot "log(lineages) should be ln (lineages)?" please confirm it.

You are correct, it is a natural logarithm of lineages in given time. We corrected the figures accordingly.

I look forward to seeing this review in print.

Reviewer #2 (Remarks to the Author):

This is a re-review and so I will keep my comments focused on the changes of the revised version. I think the authors have appropriately revised the ms from the previous round of comments.

Only one remaining part from previous -

L72 –I still think more openness is needed here about the data matrix is needed. I'd suggest 'up to 7076 bp' for example and listing the mean and variance on number of bp per MOTU.

I appreciate that the authors have increased transparency on this in the supplementary and repeated the analysis with the full matrix samples.

We included the suggested information in the manuscript. Line 76 now states:

"(7067 bp in total, with mean value per MOTU 3017 ± 2146 (standard deviation))."

I have some new/minor comments on presentation

I sort of agree with the other reviewer that it should be "bears little evidence". One could not substitute "uncovers" for bares in the sentence as written and have the grammar hold. But regardless the sentence is not good. At least it would need "The part of the world whose natural history has been most thoroughly explored has shown little evidence of the extensive ARs as found in other continents" However scientifically, I don't think this is a good sentence and it is not cited or particularly well supported. Contemporary Europe is small and temperate, and not even discrete so cannot be fairly compared to (m)any other continents. I'd suggest something like "This part of the world, whose natural history has been most thoroughly explored, has shown little evidence of ARs".

We followed the suggestion and changed the sentence (Lines 33-35): “This part of the world, whose natural history has been most thoroughly explored, has shown little evidence of extensive adaptive radiations”.

Line 46-47 is not a sentence. End with “is high”?

The original sentence stated: “Indeed, global species richness of caves and ground water peaks in European karstic areas of the Pyrenees, Southern Alps, and the Dinaric Karst.” We are not sure what reviewer had in mind. We guess that the verb *to peak* that refers to global species richness was misunderstood as a noun in plural. We used *is the highest* instead, to make sentence clearer.

Line 50: “Indeed, global species richness of caves and ground water is the highest in European karstic areas of the Pyrenees, Southern Alps, and the Dinaric Karst.”

End of line 53 should have citations of examples, I think

We added the appropriate references to the sentence (line 59) (Morvan et al. 2013, Eme et al. 2017, Lukić et al. 2019).

L 63 – it’s not clear to me if the sentence following from here is the predictions that are being tested or if the first sentence of the ms is the predictions being tested. I think the former, but being very clear at this important point is necessary. Maybe start with “Specifically, we test for ...”

We changed the whole last paragraph of the Introduction, according to Editor’s suggestions. It now states the predictions being tested, together with summary of main results. Lines 60-69:

“Here, we present and test an entirely new view on the evolution of subterranean biodiversity and on the origins of major extant European faunal components. We do so by demonstrating that *Niphargus* is a mega-diverse genus with hundreds of species that has not only evolved and diversified entirely on the European continent but has survived here from the times when the landmasses emerged from the sea (Fig. 1). We analyse the tempo and mode of diversification and ecological disparification of this exclusively subterranean clade and show that diversification patterns satisfy the predictions of evolution by adaptive radiation. Next, using spatio-temporal correlations between diversification events and the geological uplift of large carbonate massifs, we suggest that formation of caves and subterranean habitats created a multitude of ecological opportunities, the quintessential condition for adaptive radiation²⁴.”

The transition between data, background and result is not very clear. Maybe L 75 should start “We found that the genus ...”

We agree and we corrected the sentence as suggested (line 78).

L76 I’d suggest to revise to “in the region that is contemporary Western Europe”

We agree and we corrected the sentence as suggested (line 79).

L154 – the emergence of new habitats at 15 mya, independent of the findings here, was not very evident to the reader among the geological history given so far. Please refer to the figures and citation(s) needed.

We agree with the reviewer. We added the citations and referred to Figure 3, as suggested. Lines 155-157:

“In summary, all diversification analyses suggested sudden increases in the rates of species accumulation, morphological evolution and ecological disparification at approximately 15 Mya – the time when novel habitat emerged at a grand scale^{27,34,35} (Figure 3).”

No comma in line 281

We corrected the sentence.

some of the date ranges are flipping around from oldest to youngest and to youngest to oldest eg in the HPDs

We corrected the date range in line 79 so that all ranges expressed in “million years ago” are now reported from oldest to youngest.